# A tunable LIC1-adaptor interaction modulates dynein activity in a cargo-specific manner

In-Gyun Lee[1,3], Sydney E. Cason[1,2], Saif S. Alqassim [1,4], Erika L. F. Holzbaur [1,2] & Roberto Dominguez [1✉]

Cytoplasmic dynein-1 (dynein) is the motor responsible for most retrograde transport of cargoes along microtubules in eukaryotic cells, including organelles, mRNA and viruses. Cargo selectivity and activation of processive motility depend on a group of so-called "activating adaptors" that link dynein to its general cofactor, dynactin, and cargoes. The mechanism by which these adaptors regulate dynein transport is poorly understood. Here, based on crystal structures, quantitative binding studies, and in vitro motility assays, we show that BICD2, CRACR2a, and HOOK3, representing three subfamilies of unrelated adaptors, interact with the same amphipathic helix of the dynein light intermediate chain-1 (LIC1). While the hydrophobic character of the interaction is conserved, the three adaptor sub-families use different folds (coiled-coil, EF-hand, HOOK domain) and different surface contacts to bind the LIC1 helix with affinities ranging from 1.5 to 15.0 μM. We propose that a tunable LIC1-adaptor interaction modulates dynein's motility in a cargo-specific manner.

---

[1] Department of Physiology and Pennsylvania Muscle Institute, Perelman School of Medicine, University of Pennsylvania, Philadelphia, PA 19104, USA. [2] Neuroscience Graduate Group, Biomedical Graduate Studies, Perelman School of Medicine, University of Pennsylvania, Philadelphia, PA 19104, USA. [3] Present address: Korea Institute of Science and Technology (KIST), 5 Hwarangro 14-Gil, Seongbuk-Gu, Seoul 02792, Republic of Korea. [4] Present address: College of Medicine, Mohammed Bin Rashid University of Medicine and Health Sciences, Dubai, United Arab Emirates. ✉email: droberto@pennmedicine.upenn.edu

In eukaryotic cells, a single motor, cytoplasmic dynein-1 (dynein), carries out the majority of microtubule-based retrograde transport of cargoes, including membranous organelles, protein complexes, mRNA, and viruses[1]. By itself, mammalian dynein adopts an autoinhibited so-called "Phi-particle" conformation that lowers its affinity for microtubules and inhibits processive motility[2]. Dynein is activated for long-distance processive movement along microtubules through interactions with its general cofactor, dynactin, and one of many so-called "activating adaptors"[3,4]. The activating adaptors (hereafter referred to as adaptors) fulfill multiple roles; they mediate key interactions that stabilize the dynein–dynactin complex, recruit cargoes, and regulate dynein's activity and processive motility[2,3,5–9]. Both dynein (~1.4 MDa) and dynactin (~1.0 MDa) are conserved, multi-subunit complexes. The adaptors, in contrast, are generally unrelated, although they all contain long regions of coiled-coil structure and form parallel dimers. The sequence variability of the adaptors is important in at least two ways: (a) through their variable C-terminal regions, different adaptors appear to link dynein–dynactin to different cargoes[1,8,10], (b) through their variable N-terminal regions, the adaptors seem to modulate the speed, force and processivity of the dynein motor in different ways[3,9,11–13]. Here, we shed light on the structural basis for the latter mechanism.

For some of the adaptors, including BICD2, BICDL1, and HOOK3, cryo-EM structures show that a ~350 Å coiled-coil segment runs parallel to the dynactin filament of actin-related subunits and provides part of the binding interface for the dynein tails, explaining how the adaptors stabilize the dynein–dynactin complex[5,9,12]. While there is limited structural information about the variable regions N- and C-terminal to the dynactin-binding coiled-coil segments, several studies have revealed an interaction between N-terminal sequences of the adaptors and the dynein light intermediate chain-1 (LIC1)[6,14–20]. In a previous study[18], we mapped this interaction to a conserved amphipathic helix within the otherwise unstructured and poorly conserved C-terminal domain of LIC1 (human LIC1 residues 433–458). The adaptors, however, fall into at least three different subfamilies, based on their N-terminal sequences, which can contain either regions of coiled-coil, pairs of EF-hands or the HOOK domain[1]. The fact that this interaction appears to be conserved on the LIC1 side but variable on the adaptor side leads to the interesting possibility that it may serve as a regulatory mechanism, prompting different motile behaviors of the dynein motor for different cargoes.

Here, we structurally and biochemically characterize the interactions of LIC1 with two adaptors, BICD2 and CRACR2a. Together with HOOK3, examined by us previously[18], BICD2 and CRACR2a represent the three major subfamilies of LIC1–adaptor interactions studied to date. Crystallographic analysis shows that the LIC1–adaptor interaction has a conserved hydrophobic character, with the amphipathic LIC1 helix displaying a nearly identical conformation and its hydrophobic face embedded in a hydrophobic cleft in the three adaptor subfamilies. Yet, on the adaptor side, the interaction is entirely different, involving different folds (coiled-coil, EF-hand, and HOOK domain) and different protein–protein contacts in the three adaptor subfamilies. Quantitative analysis of the interactions using isothermal titration calorimetry (ITC) reveals their dimeric character and affinities ranging from 1.5 to 15.0 μM. The $Ca^{2+}$-dependence and functional role of the CRACR2a–LIC1 interaction was confirmed in the crystal structure, as well as in binding studies and in vitro motility assays. In contrast, other EF-hand containing adaptors (FIP3 and NIN) bind the LIC1 helix in a $Ca^{2+}$-independent manner. Together, the results reveal a tunable LIC1–adaptor interaction that may modulate the force, speed, and processivity of the dynein motor.

## Results

**LIC1 interaction with CC1-box-containing dynein–dynactin adaptors.** A subfamily of dynein–dynactin adaptors, including BICD1–2 and BICDL1–2, Spindly, TRAK1–2 and HAP1, share a coiled-coil segment known as the CC1-box that has been directly implicated in LIC1 binding for some of these proteins[16,18,19] (Supplementary Table 1). We previously found that fragments encompassing the CC1-box of two of these proteins, $BICD2_{1–98}$ and $Spindly_{1–142}$, were dimeric and bound two LIC1 helices ($LIC1_{433–458}$) with dissociation constants in the low-micromolar range[18]. Here, we set out to determine the structural basis of this interaction. A 2.4-Å resolution structure of human $BICD2_{1–98}$ in complex with human $LIC1_{433–458}$ was determined ab initio using the single-wavelength anomalous dispersion method from crystals of selenomethionine-substituted $BICD2_{1–98}$ (Table 1). The electron density map reveals residues P4–A81 of one chain and L13–E80 of the other chain of BICD2, and residues E441–L452 of a single LIC1 helix (Fig. 1a, b). The uninterrupted coiled-coil segment of BICD2 comprises residues P19–L78, displaying the characteristic heptad repeat (Fig. 1c). Coiled-coil prediction programs also suggest a break of the coiled-coil structure around P19 and E80, consistent with the presence of typical helix-breaking amino acids at these two locations (P19 and G83). At the N-terminus, one of the BICD2 chains is disordered, whereas the other chain contains an additional helix (residues E6–E16) that runs antiparallel to the coiled-coil helices. This type of fold is

**Table 1 Data collection and refinement statistics.**

| Data collection (Se-Met derivative) | BICD2-LIC1 complex[a] | CRACR2a-LIC1 complex[a] |
|---|---|---|
| Wavelength (Å) | 0.9775 | 0.9793 |
| X-ray source | MacCHESS beamline F1 | SSRL beamline 14-1 |
| Space group | $P\,2_1\,2_1\,2_1$ | $P\,2_1$ |
| Cell dimensions | | |
| $a, b, c$ (Å) | 54.77, 61.84, 81.68 | 57.29, 161.99, 56.90 |
| $\alpha, \beta, \gamma$ (°) | 90.0, 90.0, 90.0 | 90.0, 104.01, 90.0 |
| Resolution (Å) | 49.04–2.40 | 43.35–2.66 |
| | (2.49–2.40)[b] | (2.78–2.66) |
| $R_{merge}$ | 0.109 (0.785) | 0.359 (0.953) |
| $R_{pim}$ | 0.033 (0.289) | 0.114 (0.330) |
| $I\,/\,\sigma(I)$ | 12.0 (1.8) | 7.2 (3.0) |
| No. of reflections | 218993 (8766) | 303250 (27481) |
| No. of unique reflections | 19602 (1396) | 28181 (2798) |
| Completeness (%) | 94.9 (67.1) | 99.9 (100.0) |
| Redundancy | 11.2 (6.3) | 10.8 (9.8) |
| Wilson B-factor (Å$^2$) | 35.8 | 33.3 |
| $CC_{1/2}$ | 0.99 (0.76) | 0.96 (0.44) |
| Effective resolution[c] | | |
| Overall d_min (Å) | 2.40 | 2.50 |
| d_min along $a^*, b^*, c^*$ (Å) | 2.88, 2.57, 2.47 | 2.52, 3.00, 2.50 |
| Refinement | | |
| Resolution (Å) | 34.08–2.40 | 43.35–2.66 |
| | (2.49–2.40) | (2.73–2.66) |
| No. of unique reflections | 14,423 (907) | 23,908 (980) |
| Completeness (%)[d] | 69.2 (23.0) | 94.5 (50.6) |
| $R_{work}/R_{free}$ (%) | 19.2 (26.8)/24.6 (32.5) | 19.7 (25.5)/23.2 (30.6) |
| No. of atoms | | |
| Protein | 1309 | 5543 |
| $Ca^{2+}$ | – | 8 |
| Water | 42 | 49 |
| B-factors (Å$^2$) | | |
| Protein | 49.8 | 63.8 |
| $Ca^{2+}$ | – | 49.5 |
| Water | 38.1 | 42.1 |
| R.m.s. deviations | | |
| Bond lengths (Å) | 0.008 | 0.017 |
| Bond angles (°) | 0.98 | 1.50 |
| Ramachandran (%) | | |
| Favored | 96.0 | 95.60 |
| Outliers | 0.6 | 0.0 |
| PDB code | 6PSE | 6PSD |

[a]Merged dataset from two crystals.
[b]Values in parentheses correspond to the highest resolution shell.
[c]Calculated with the program Phenix.xtriage[44].
[d]Anisotropy data correction with HKL2000 eliminated weak, unreliable reflections, while reducing the completeness of the data used in refinement.

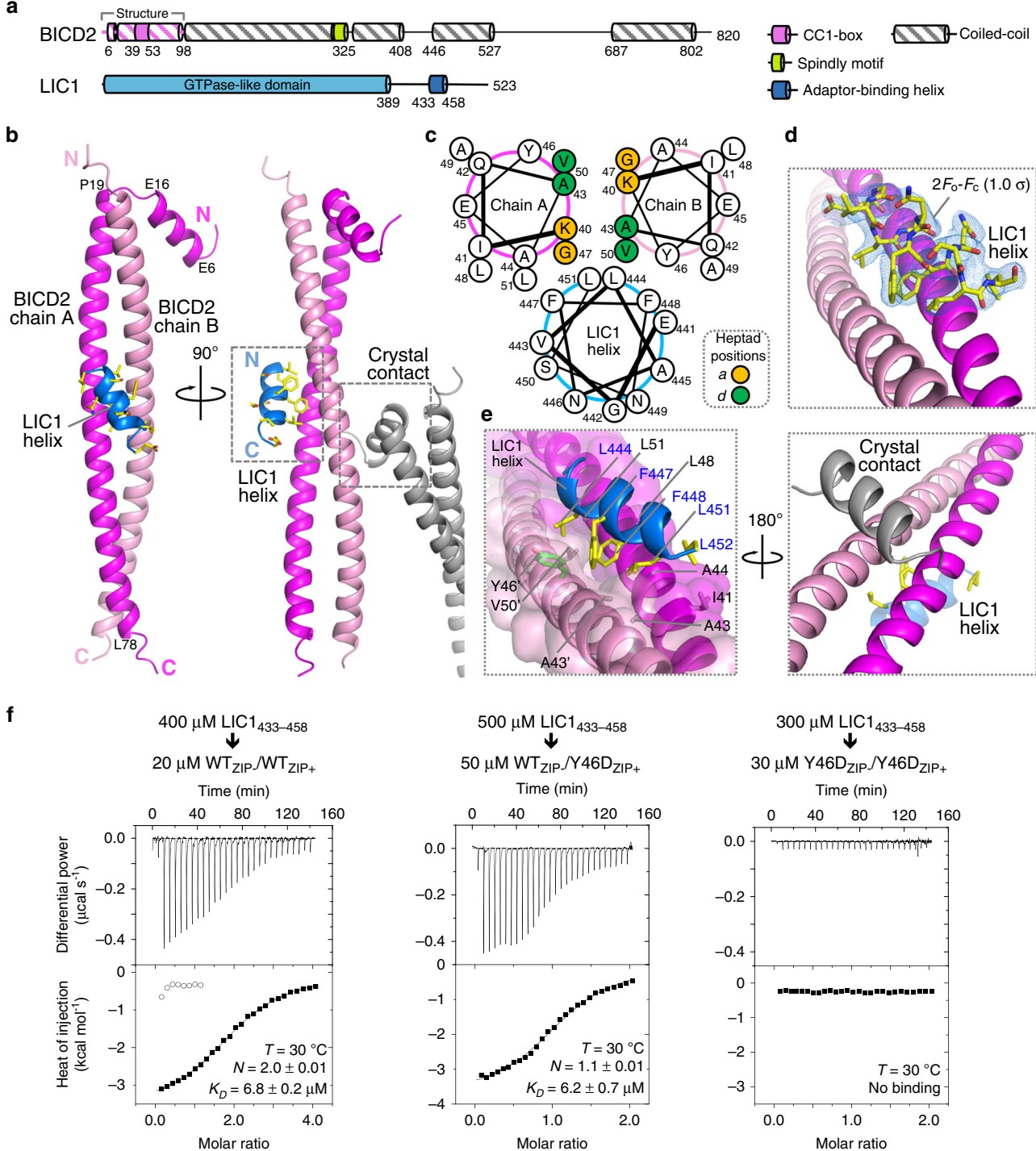

**Fig. 1 LIC1 interaction with CC1-box-containing dynein–dynactin adaptors. a** Domain organization of human LIC1 and BICD2, highlighting the BICD2$_{1-98}$ fragment containing the CC1-box that was co-crystallized with LIC1$_{433-458}$. **b** Ribbon diagram showing two perpendicular views of the structure of the complex of BICD2$_{1-98}$ (coiled-coil chains A and B are colored magenta and pink, respectively) with LIC1$_{433-458}$ (blue). The side chains of LIC1$_{433-458}$ are shown using a stick representation and colored by atom type. A symmetry-related BICD2$_{1-98}$ molecule (gray) occludes the second LIC1$_{433-458}$ binding site. **c** Helical-wheel representation of the interaction of LIC1$_{433-458}$ with the CC1-box motif of BICD2. **d**, **e** Close-up views showing the interactions of LIC1$_{433-458}$ and a symmetry-related molecule on opposite sides of the BICD2 coiled-coil. The 2Fo–Fc electron density map (blue mesh) contoured at 1.0σ is shown around an all-atom representation of the LIC1 helix. Note that residues on the hydrophobic face of the LIC1 helix insert into a hydrophobic pocket formed at the interface of the two BICD2 chains. BICD2 residue Y46, which was mutated in this study, is shown in green. **f** ITC titrations of LIC1$_{433-458}$ into the indicated heterodimers of WT and mutant Y46D BICD2 obtained by fusing a ZIP+/ZIP− leucine zipper C-terminally to the BICD2 constructs (see also Supplementary Fig. 1). ITC data shown are representative of $n=3$ independent experiments, with similar results. Listed for each titration are the experimental conditions and fitting parameters, including the dissociation constant ($K_D$) and binding stoichiometry ($N$) derived from fitting to a binding isotherm (black line). Open symbols correspond to a control titration into buffer, shown for the first eight injections of the first experiment, since all the proteins were dialyzed in parallel. The $K_D$ and $N$ values are given as mean ± s.d. of the fits.

somewhat reminiscent of the Rab-interacting lysosomal protein (RILP) homology-1 (RH1) domain in the structure of RILPL2 (RILP-like protein 2) bound to the globular tail domain of myosin-V[21]. Both structures consist of a central coiled-coil preceded by a helix that runs antiparallel to the coiled-coil. As its name indicates, RILPL2 is related to RILP, which together with JIP1–4 (JNK-interacting proteins 1–4) forms part of a subfamily of RH1 domain-containing proteins[22] that has been implicated in interactions with the C-terminal region of LIC1[17,23], although these proteins are not confirmed dynein–dynactin adaptors[1].

The LIC1 helix binds near the middle of the BICD2 coiled-coil segment and is oriented diagonally with respect to the axis of the coiled-coil (Fig. 1b). Like in the structure of the LIC1 helix bound to the HOOK domain[18], residues L444, F447, F448, and L451 on the hydrophobic face of this helix insert into a hydrophobic pocket formed at the interface between the two chains of BICD2 (Fig. 1c–e). Critical to the formation of this pocket is the presence of highly conserved alanine and glycine residues in the CC1-box that, based on the structure, can be more precisely defined as consisting of human BICD2 residues $^{39}$EKIQ**AA**EY**G**LA**VL**EE$^{53}$ (where, bold letters indicate residue conservation ≥50% among 100 CC1-box-containing sequences, Supplementary Fig. 1a). The sequence of the CC1-box is atypical for a coiled-coil structure. In particular, residues A43 and G47 occupy "d" and "a" positions within the hydrophobic core of the heptad repeat and, because of their unusually small size for these positions, they help create a cavity for the binding of the LIC1 helix (Fig. 1c, e).

The presence of a single LIC1 helix bound to BICD2 in the structure is surprising (Fig. 1b, d), since previous analysis suggested two binding sites[18]. However, the second LIC1-binding site, predicted to be symmetrically positioned on the opposite side of the BICD2 coiled-coil from the one observed in the structure, is masked by a crystal packing contact (Fig. 1b, e). Therefore, to conclusively answer whether BICD2 can simultaneously bind two LIC1 helices we made BICD2 heterodimers, consisting of different combinations of a WT subunit and a subunit carrying the mutation Y46D in the middle of the CC1-box, which is predicted to disrupt LIC1 binding (Fig. 1e). To ensure homogeneous heterodimer formation during coexpression, we appended a heterodimerizing leucine zipper (ZIP+/ZIP−)[24] C-terminally to the BICD2 constructs (Supplementary Fig. 1b). Heterodimer formation was confirmed by sodium dodecyl sulfate polyacrylamide gel electrophoresis and light scattering analysis (Supplementary Fig. 1c, d). Three variants of this construct were made, a half-functional heterodimer in which only one BICD2 chain was mutated (Y46D$_{ZIP+}$/WT$_{ZIP-}$) and two controls, Y46D$_{ZIP+}$/Y46D$_{ZIP-}$ and WT$_{ZIP+}$/WT$_{ZIP-}$, which should not bind the LIC1 helix or bind two helices, respectively. Binding was quantitatively assessed using ITC. Consistent with our previous observations[18], the titration of LIC1$_{433–458}$ into WT$_{ZIP+}$/WT$_{ZIP-}$ produced an exothermic reaction that fitted best to a two-site binding isotherm ($N = 2.0$) with $K_D \sim 6.8\,\mu M$ for each of the sites (Fig. 1f and Supplementary Fig. 1e). As anticipated, the titration of LIC1$_{433–458}$ into Y46D$_{ZIP+}$/Y46D$_{ZIP-}$ was undistinguishable from a control titration into buffer, i.e., no heat of exchange was observed, indicating lack of binding. In contrast, Y46D$_{ZIP+}$/WT$_{ZIP-}$ bound a single LIC1 helix ($N = 1.1$) with $K_D \sim 6.2\,\mu M$ (Fig. 1f). These results conclusively show that BICD2 and likely other CC1-box-containing adaptors have the ability to interact with two LIC1 helices and with similar low-micromolar affinities.

**LIC1 interaction with EF-hand-containing dynein–dynactin adaptors.** CRACR2a (calcium release-activated calcium channel regulator 2A), Rab45 (Ras-related protein Rab-45), FIP3 (Rab11

family-interacting protein 3), and NIN (Ninein) form part of a subfamily of dynein–dynactin adaptors characterized by the presence of EF-hand domains[3,8,10] (Fig. 2a and Supplementary Table 1). In pull-down experiments, FIP3 and NIN have been shown to interact with the LIC1 helix[19], although the interaction was not specifically mapped to the EF-hand domains. Yet, for at least one of these adaptors, CRACR2a, the ability to activate processive dynein–dynactin motility is regulated by Ca$^{2+}$,[10], which indirectly implicates the EF-hand domains. The EF-hand domain is a protein–protein interaction module that almost invariably functions as pairs of EF-hand domains and is often, albeit not always, regulated by Ca$^{2+}$ binding to either one or both EF-hands in a pair[25]. We set out to directly test whether the LIC1 helix binds to the EF-hand domains of CRACR2a, FIP3 and NIN, and whether Ca$^{2+}$ regulates the interactions.

CRACR2a and FIP3 each contains a single EF-hand pair, whereas NIN contains two EF-hand pairs in addition to a single EF-hand domain that precedes the first coiled-coil segment (Fig. 2a). Therefore, in these experiments we used constructs comprising individual EF-hand pairs of the three proteins (MBP-CRACR2a$_{47–122}$, FIP3$_{206–270}$, and NIN$_{1–87}$) and construct NIN$_{180–356}$, encompassing the last three EF-hand domains of NIN. Binding of these adaptors to either full-length LIC1 (construct MBP-LIC1$_{FL}$) or the LIC1 helix (LIC1$_{433–458}$) was measured using ITC, both in the presence and the absence of CaCl$_2$. CRACR2a$_{47–122}$ has low UV-absorbance, and thus to facilitate the determination of accurate protein concentrations and stoichiometries for fitting of the ITC experiments the N-terminal maltose-binding protein (MBP) tag was not removed. To maximize the heat of injection signal of each titration and thus the accuracy of the fits, the temperature was set to either 20 or 30 °C.

In the presence of 5 mM CaCl$_2$, MBP-CRACR2a$_{47–122}$, FIP3$_{206–270}$, and NIN$_{1–87}$ bound MBP-LIC1$_{FL}$ with low-micromolar affinities ($K_D$s ~ 3.6, 6.1, and 12.0 μM, respectively) and ~1:1 stoichiometry (Fig. 2b). In contrast, in the presence of 5 mM EGTA (a Ca$^{2+}$ chelator) MBP-CRACR2a$_{47–122}$ failed to bind MBP-LIC1$_{FL}$, whereas the binding of FIP3$_{206–270}$ and NIN$_{1–87}$ was mostly unaffected (Fig. 2c). We separately confirmed that Ca$^{2+}$ does not bind to MBP (Supplementary Fig. 2a), such that the Ca$^{2+}$-dependance of the interaction between MBP-CRACR2a$_{47–122}$ and MBP-LIC1$_{FL}$ must be contained within the EF-hands of CRACR2a$_{47–122}$. We also established that FIP3$_{206–270}$ and NIN$_{1–87}$ are properly folded in the absence of Ca$^{2+}$, displaying circular dichroism spectra characteristic of α-helical structures, with minima around 222 and 208 nm and no evidence of unfolding (Supplementary Fig. 2b). Finally, NIN$_{180–356}$ did not bind MBP-LIC1$_{FL}$, suggesting that the three EF-hand domains contained within this construct are not implicated in the LIC1 interaction (Supplementary Fig. 2c). While Ca$^{2+}$ is present during expression/purification, neither CaCl$_2$ nor EGTA were added in the latter experiment, since the EF-hands of NIN$_{180–356}$ lack key residues necessary for Ca$^{2+}$ binding (see below).

For other subfamilies of adaptors, we have found that the amphipathic helix LIC1$_{433–458}$ fully accounts for the LIC1-adaptor interaction (Fig. 1f)[18]. Consistently the three EF-hand-containing adaptors analyzed here bound this helix with the same stoichiometry and somewhat higher affinity as full-length LIC1 (Fig. 2d). The LIC1 helix may be partially occluded within construct MBP-LIC1$_{FL}$, which could explain why the affinity of the isolated helix is consistently higher for the three adaptors. The affinities of Ca$^{2+}$-CRACR2a ($K_D$ ~ 1.4 μM), FIP3 ($K_D$ ~ 2.0 μM) and NIN ($K_D$ ~ 5.4 μM) for the LIC1 helix fall within the same range as those previously measured by us for BICD2 ($K_D$ ~ 1.5–7.5 μM), Spindly ($K_D$ ~ 3.5–7.6 μM), HOOK1 ($K_D$ ~ 15.7 μM), and HOOK3 ($K_D$ ~ 6.3 μM)[18].

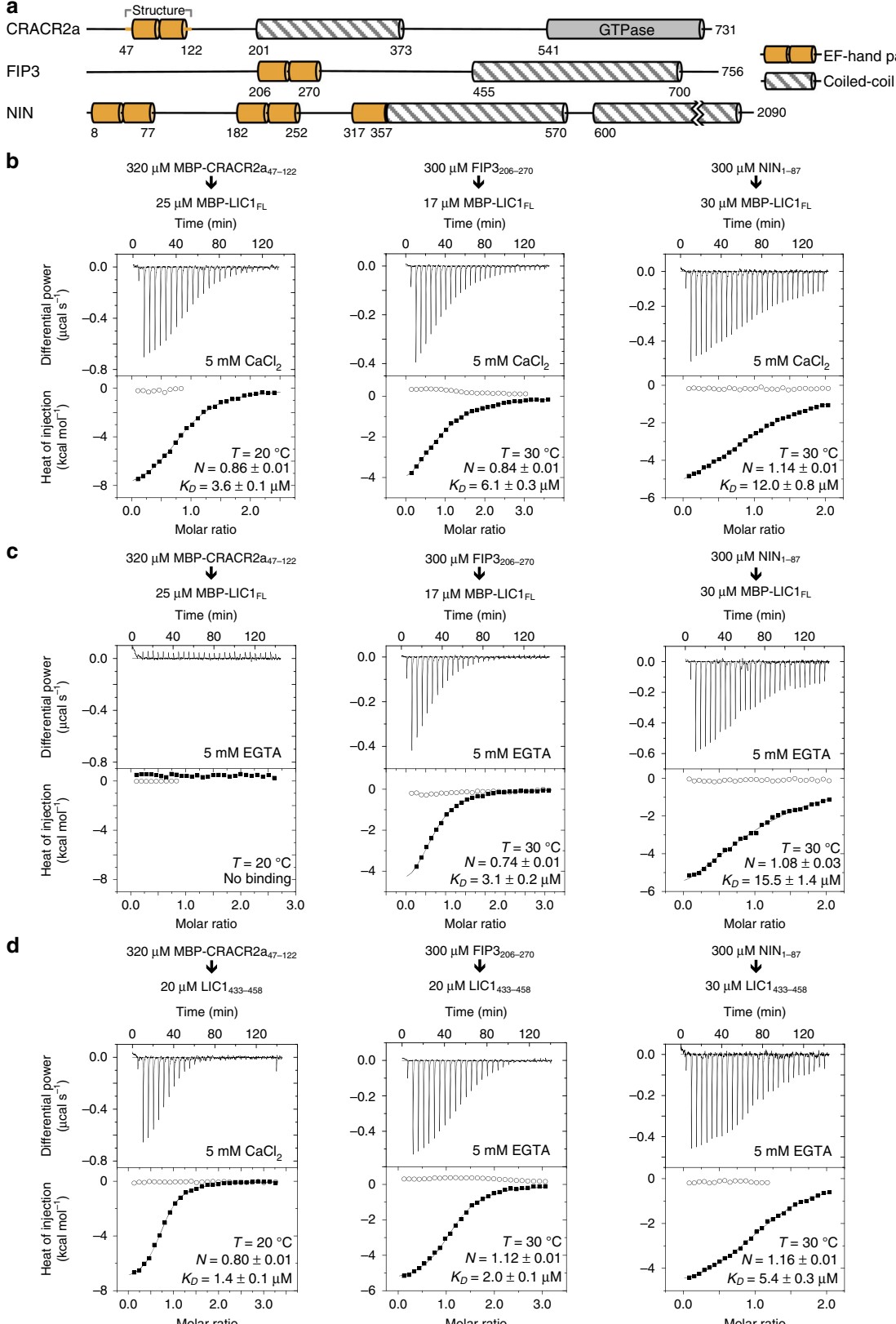

**Fig. 2 LIC1 interaction with EF-hand-containing dynein–dynactin adaptors. a** Domain organization of EF-hand-containing adaptors, highlighting the EF-hand pair of CRACR2a that was co-crystallized with LIC1$_{433-458}$. **b–d** ITC titrations of EF-hand pairs from human CRACR2a, FIP3 and NIN into MBP-LIC1$_{FL}$ or LIC1$_{433-458}$ and in the presence of 5 mM CaCl$_2$ or 5 mM EGTA (as indicated). ITC data shown are representative of $n = 2$ independent experiments, with similar results. Listed for each titration are the experimental conditions and fitting parameters, including the dissociation constant ($K_D$) and binding stoichiometry ($N$) derived from fitting to a binding isotherm (black line). The $K_D$ and $N$ values are given as mean ± s.d. of the fits. Open symbols correspond to a control titration into buffer.

These results suggest that the EF-hand pair of CRACR2a is regulated by $Ca^{2+}$, whereas those of FIP3 and NIN are not. To further explore the $Ca^{2+}$-dependence of the LIC1 interactions we assessed the ability of these EF-hand pairs to bind $Ca^{2+}$ by ITC. The titration of $CaCl_2$ into MBP-CRACR2a$_{47-122}$ revealed a single, relatively low-affinity $Ca^{2+}$-binding site ($N \sim 0.90$; $K_D \sim 14.7\,\mu M$), whereas FIP3$_{206-270}$ and NIN$_{1-87}$ did not bind $Ca^{2+}$ (Fig. 3a). To understand the source of these differences, we compared the sequences of the EF-hands of these three proteins to those of 3071 EF-hand domains from known $Ca^{2+}$-binding proteins (Fig. 3b). This analysis shows that amino acids crucial for $Ca^{2+}$-binding in these proteins (conservation score $\geq 0.5$) are conserved in the second EF-hand of CRACR2a and in both EF-hands of Rab45, but several are mutated in the first EF-hand of CRACR2a and in the EF-hands of FIP3 and NIN (Fig. 3b).

**Crystal structure of the EF-hand pair of CRACR2a in complex with the LIC1 helix**. A crystal structure of CRACR2a$_{47-122}$ (MBP removed) in complex with LIC1$_{433-458}$ was obtained at 2.66 Å resolution (Fig. 4a and Table 1). The structure was determined ab initio using the single-wavelength anomalous dispersion method from crystals of selenomethionine-substituted CRACR2a$_{47-122}$. The monoclinic crystals, belonging to space group P2$_1$, contain eight complexes in the asymmetric unit. Two of the complexes are poorly defined in the electron density map due to higher mobility, yet eightfold non-crystallographic symmetry averaging was used throughout the initial steps of the refinement, yielding well-defined electron density for CRACR2a residues G47–Q120 and LIC1 residues T439–K454 (Fig. 4b, c). The non-crystallographic symmetry restraints were then relaxed toward the end of the refinement, resulting in RMSDs of <0.4 Å for equivalent Cα atoms among the eight complexes in the asymmetric unit (Supplementary Fig. 3a).

The two EF-hands of CRACR2a display a canonical fold, consisting of incoming helix (helices α1 and α3), loop and exiting helix (helices α2 and α4) (Figs. 3b and 4a). As indicated by the ITC titration (Fig. 3a), the structure reveals a single $Ca^{2+}$-binding site. Consistent with the sequence analysis (Fig. 3b), the $Ca^{2+}$ density is observed within the second EF-hand domain of all the complexes of the asymmetric unit (Fig. 4d), whereas none of the complexes displays $Ca^{2+}$ density within the first EF-hand (Supplementary Fig. 3b). The conformation and $Ca^{2+}$ coordination of the second EF-hand of CRACR2a is similar to those of troponin C and calmodulin (Fig. 4d). $Ca^{2+}$ is most commonly coordinated by six ligands that form a sevenfold, pentagonal–bipyramidal ligation network around the ion, with bond distances ranging from 2.3 to 2.6 Å[25,26]. In CRACR2a, $Ca^{2+}$ is ligated by the side chains of D97, D99, N101, E108, the main-chain carbonyl oxygen of Y103 and a water molecule that is in turn coordinated by T105 (Fig. 4d). In the traditional $Ca^{2+}$-binding loop nomenclature[25], these amino acids occupy positions X (D97), Y (D99), Z (N101), −Y (Y103), −X (T105), and −Z (E108) (Fig. 3b). Like in other EF-hands that bind $Ca^{2+}$, the amino acid at position −Z coordinates the $Ca^{2+}$ in a bidentate fashion to complete the sevenfold ligation network. To understand why certain residues are allowed at specific positions of the $Ca^{2+}$-binding loop while others are not, we need to analyze the structures. Thus, positions Y and Z can be either D, N, or S because these positions contribute a single side-chain oxygen to the interaction with $Ca^{2+}$ (Figs. 3b and 4d). Similarly, position −X contributes a main chain contact, and accordingly almost any amino acid is allowed at this position. In contrast, only E is allowed at position −Z, and not even D, which has a shorter side chain, can account for this bidentate interaction (see "Discussion").

Like in other members of the EF-hand family, the target-binding site consists of a hydrophobic cleft formed between the incoming and exiting helices of the two EF-hand domains. The hydrophobic face of the LIC1 helix binds in this cleft (Fig. 4a–c), and the same conformation is observed in all eight complexes of the asymmetric unit (Supplementary Fig. 3a). Like in the complexes with the HOOK domain[18] and BICD2 (Fig. 1e), highly conserved, hydrophobic residues of the LIC1 helix (L444, F447, F448, and L451) insert into the hydrophobic cleft formed by the EF-hand pair of CRACR2a (Fig. 4c). Because CRACR2a does not bind LIC1 in the absence of $Ca^{2+}$ (Fig. 2c), it is likely that this cleft only opens in the $Ca^{2+}$-bound state, which is a common feature among EF-hand proteins[25]. However, other EF-hands, including those in some myosin light chains[27], open in the presence of a target sequence, without the need for $Ca^{2+}$ pre-activation, as appears to be the case for FIP3 and NIN. A sequence alignment of EF-hand pairs that either bind or do not bind the LIC1 helix fails to reveal specific amino acids that would be present in one group and not the other, which based on the limited information we have now precludes predicting potential interactions of LIC1 with other EF-hands (Supplementary Fig. 3c).

**Disrupting the CRACR2a–LIC1 interaction impairs dynein motility in vitro**. We disrupted the CRACR2a–LIC1 interaction through mutagenesis to assess its functional importance in CRACR2a-induced dynein–dynactin motility using a single-molecule assay. CRACR2a residue F58 in the hydrophobic cleft that binds the LIC1 helix was mutated to aspartic acid and the mutant's inability to bind the LIC1 helix in the presence of $Ca^{2+}$ was confirmed by ITC (Fig. 5a). We then used total internal reflection fluorescence (TIRF) microscopy to track the movement of CRACR2a–dynein–dynactin complexes from lysates of cells expressing either wild-type (WT) or mutant F58D TMR-labeled Halo-CRACR2a and in the presence of either 2 μM free $Ca^{2+}$ or 2 mM EGTA (Fig. 5b and Supplementary Fig. 4). Polarity-marked dynamic microtubules (HiLyte Fluor 488-labeled microtubules growing from HiLyte Fluor 647-labeled seeds) were used in these experiments to ensure only retrograde, dynein-driven runs (accounting for ~95% of all the runs) were analyzed (Supplementary Fig. 4c). For WT CRACR2a, the number of processive motile events, defined as runs of ≥1.0 μm, was significantly higher with than without calcium, and the F58D mutation reduced the number of runs to that seen with WT CRACR2a in the absence of calcium (Fig. 5c). No runs were observed when HaloTag was expressed alone as a negative control (Supplementary Fig. 4d). In the presence of 2 μM free $Ca^{2+}$, motile events with WT CRACR2a had a mean velocity of 2 μm s$^{-1}$ (Fig. 5d), consistent with in vivo measurements of dynein velocity[28]. The limited number of runs observed with the mutant in the presence of calcium displayed a similar mean velocity (Fig. 5d). However, motile events were significantly shorter with the mutant than with WT CRACR2a (Fig. 5e). These results are generally consistent with a recent study[10] and confirm the importance and $Ca^{2+}$-dependence of the CRACR2a–LIC1 interaction for dynein–dynactin processive motility.

## Discussion

Contrary to kinesins that form a large superfamily with specialized functions and dedicated cargoes, long-range retrograde transport of most cargoes along microtubules, including RNAs, cellular organelles, and viruses, is driven by a single motor, dynein. For dynein, regulation of the motor's activity depends on numerous intramolecular and intermolecular factors, including posttranslational modification of the microtubule track[29],

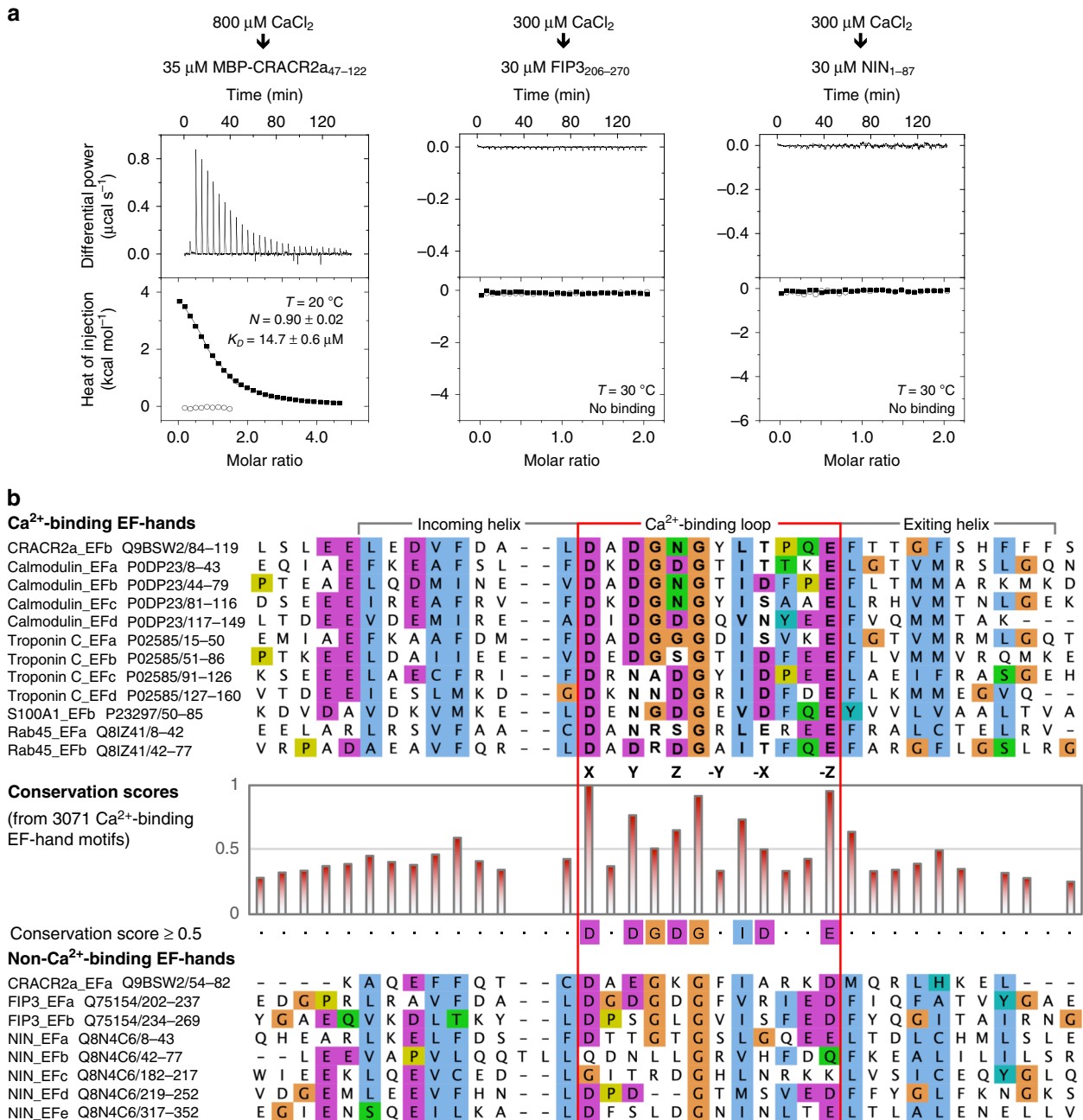

**Fig. 3 Assessing the ability of EF-hand pairs of dynein–dynactin adaptors to bind Ca²⁺.** **a** ITC titrations of CaCl₂ into EF-hand pairs from human CRACR2a, FIP3, and NIN (as indicated). ITC data shown are representative of $n = 2$ independent experiments, with similar results. Listed for each titration are the experimental conditions and fitting parameters, including the dissociation constant ($K_D$) and binding stoichiometry ($N$) derived from fitting to a binding isotherm (black line). The $K_D$ and $N$ values are given as mean ± s.d. of the fits. Open symbols correspond to a control titration into buffer. **b** Comparison of the sequences of EF-hand pairs of CRACR2a, FIP3, NIN, and Rab45 with those of classical Ca²⁺-binding EF-hands from calmodulin (four EF-hands), skeletal muscle troponin C (four EF-hands), and S100A1 (one high-affinity Ca²⁺-binding EF-hand). The sequences are all human and UniProt accession codes are listed with the name of each sequence. The adaptors are separated into Ca²⁺-binding (CRACR2a and Rab45) and Non-Ca²⁺-binding (FIP3 and NIN). This classification is based on the experimental results shown in part **a** and the conservation of key amino acids involved in the coordination of Ca²⁺, with per-residue conservation scores ≥50% among 3071 sequences of known Ca²⁺-binding EF-hands (middle graph), selected using PROSITE (http://prosite.expasy.org/) with matrix PS50222 and pattern PS00018. According to the classical nomenclature[25], amino acids implicated in the pentagonal bipyramidal coordination of Ca²⁺ occupy positions X, Y, Z, −Y, −X, and −Z (as indicated). See main text for why certain Ca²⁺ coordinating positions are highly conserved whereas others can vary.

microtubule-associated proteins[30,31], cargo binding[32], auto-inhibitory interactions[2,9,33], and dynein-binding proteins such as LIS1 and NudE/NudEL[34–37]. In addition, a large family of activating adaptors provides a physical link with specific cargoes and dynein's dedicated cofactor, dynactin, while also contributing to activation of processive motility[1–4,12,13] (Fig. 6a–c and Supplementary Table 1). Although these adaptors invariably contain long regions of coiled-coil structure, and thus form parallel

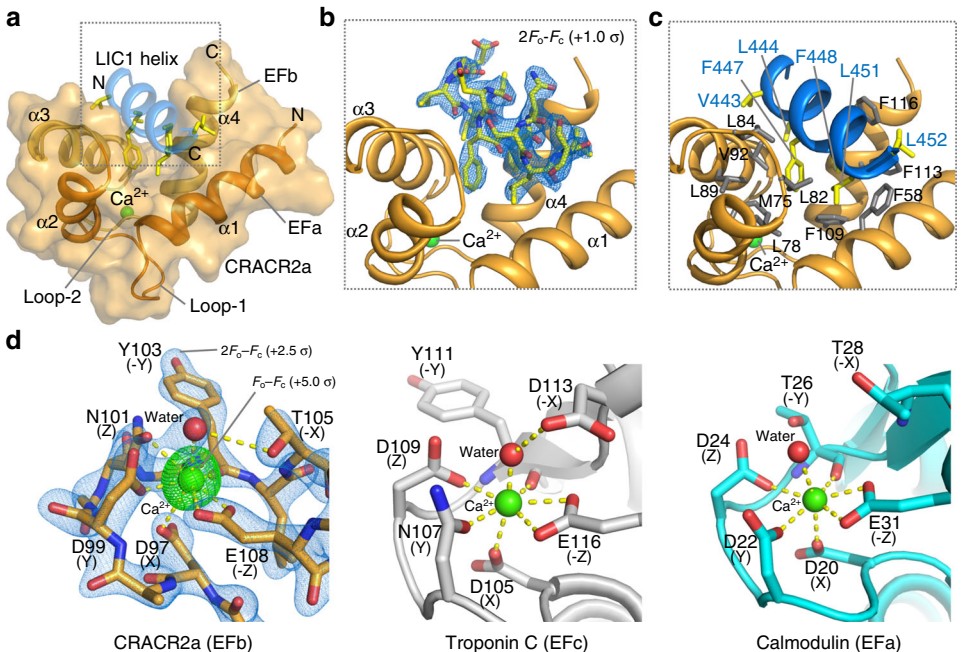

**Fig. 4 Crystal structure of the EF-hand pair of CRACR2a in complex with LIC1_{433–458}. a** Ribbon diagram and surface representation of the structure of the complex of human CRACR2a_{47–122} (the first and second EF-hands are colored dark and light orange, respectively) with LIC1_{433–458} (blue). The side chains of LIC1_{433–458} are shown using a stick representation and colored by atom type. **b, c** Close-up views showing the binding of LIC1_{433–458} in a hydrophobic cleft formed between the incoming (α1 and α3) and exiting (α2 and α4) helices of the two EF-hands. The 2Fo–Fc electron density map (blue mesh) contoured at 1.0σ is shown around an all-atom representation of the LIC1 helix. Note that residues on the hydrophobic face of the LIC1 helix insert into the hydrophobic cleft formed by the EF-hand pair (side chains colored gray). **d** Close-up view of the Ca$^{2+}$-binding loop (located within the second EF-hand) of human CRACR2a (left) compared to the third Ca$^{2+}$-binding loop of human troponin C (middle, PDB code: 1J1D) and the first Ca$^{2+}$-binding loop of human calmodulin (right, PDB code: 1CLL). The 2Fo–Fc electron density map (blue mesh) contoured at 2.5σ is shown around the atoms of CRACR2a and a Fo–Fc difference map (green mesh) contoured at 5.0σ is shown around the bound Ca$^{2+}$ ion.

dimers, they are generally unrelated at the sequence level. Because folds and mechanisms are typically more highly conserved than sequences, the current study of BICD2 and CRACR2a, together with our previous analysis of HOOK-family members[18] and cryo-EM investigations of adaptor–dynein–dynactin complexes[5,9,12], allow us to draw some general conclusions about the mechanism of action of the adaptor family.

It is informative to use the interaction with dynactin as a reference to compare the various adaptors. Dynactin consists of a central actin-like filament, containing eight actin-related protein-1 (Arp1) subunits and one β-actin subunit, and capped at the barbed end by a CapZαβ heterodimer and at the pointed end by a complex of four subunits (Arp11, p62, p25, and p27)[5]. Three other subunits (p150^{Glued}/p135, p50 and p24) form the so-called shoulder domain that binds asymmetrically on one side of the dynactin filament (Fig. 6d). Cryo-EM studies of three adaptors (BICD2, BICDL1, and HOOK3) suggest that they all contain a central ~350 Å coiled-coil segment, corresponding to ~270 amino acids, that runs along the entire length of the dynactin complex[5,9,12]. The adaptors bind approximately opposite to the shoulder domain, with their C-termini directed toward the pointed end of the dynactin filament. The adaptors provide part of the binding interface for the dynein tails on dynactin, thus helping to bring together the >2.5 MDa adaptor–dynein–dynactin complexes[5,9,12]. Despite these shared features, the dynactin-binding coiled-coil segments of the various adaptors differ in sequence and do not occupy exactly the same position at the dynein–dynactin interface[9]. Moreover, some adaptors may favor the recruitment of a single dynein to dynactin (e.g., BICD2), whereas others preferentially recruit two dyneins (e.g., BICDL1 and HOOK3)[9,12]. Therefore, although structurally conserved, the

coiled-coil segments are not interchangeable among adaptors, and provide a first level of regulation of the dynein motor.

Marking the end of the dynactin-binding coiled-coil segment, the adaptors have a short sequence known as the Spindly motif[16], which contacts subunits p25/p27 at the pointed end of the dynactin complex[5,9]. The Spindly motif appears to be the sole element of the sequence conserved among most (possibly all) the adaptors (Fig. 6a–d and Supplementary Fig. 5), providing a useful reference to map their interactions with dynactin. Less is known about the sequences C-terminal to the Spindly motif, which diverge the most among the adaptors and are thought to link dynein-dynactin directly or indirectly to specific cargoes (Supplementary Table 1), somewhat analogous to the variable tail domains of the kinesin family.

Dynein–dynactin interactions mediated by the Spindly motif and the coiled-coil segment of the adaptors are insufficient to promote the activation of dynein's motility, i.e., the transition from the auto-inhibited Phi-particle conformation to the open and parallel conformations that have higher affinity for both microtubules and dynactin[2,33]. An additional layer of regulation involves the interaction between a conserved amphipathic helix within the low-complexity C-terminal region of LIC1 and structurally diverse domains N-terminal to the dynactin-binding coiled-coil segment of the adaptors[6,16,18,19]. Here, based on structural and biochemical analysis of this interaction, we classify the adaptors into three subfamilies: CC1-box-, EF-hand-, and HOOK domain-containing adaptors (Fig. 6a–c and Supplementary Table 1).

Interestingly, while the LIC1–adaptor interaction has a conserved hydrophobic character and is conserved on the LIC1 side, it is notably variable on the side of the adaptors (Supplementary

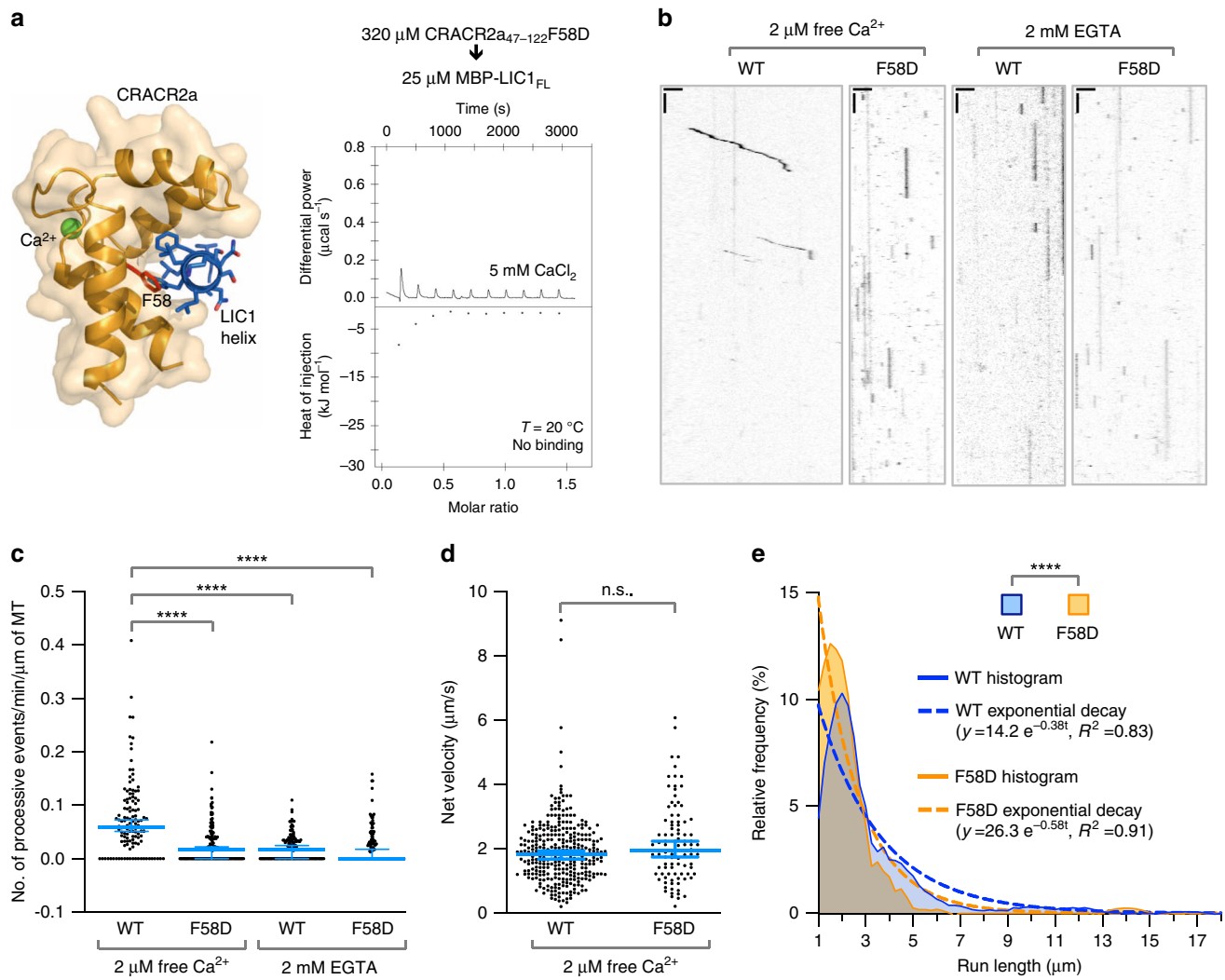

**Fig. 5 Disrupting the CRACR2a–LIC1 interaction impairs dynein motility. a** Mutating residues F58 (red) in the hydrophobic cleft of CRACR2a (orange) that bids the LIC1 helix (blue) to aspartic acid inhibits this interaction, as revealed by the ITC titration of CRACR2a$_{47-122}$ F58D into MBP-LIC1$_{FL}$ (experimental conditions given in the figure). The lack of change in the heat of injection precludes fitting to a binding isotherm. **b** Representative kymographs of the dynein–dynactin driven motility of single Halo-CRACR2a-positive particles (WT and mutant F58D CRACR2a) analyzed by TIRF microscopy with either 2 µM free Ca$^{2+}$ (left) or 2 mM EGTA (right). Kymographs are oriented with the + end of the microtubule (MT) on the left (scale bars: horizontal, 3 µm; vertical, 5 s). **c** Quantitative analysis of the number of processive motile events, defined as runs of ≥1 µm, observed under the conditions shown in part **b** of this figure. Each point represents a single microtubule ($n = 123$ per condition). The statistical significance of the measurements was determined using the Kruskal–Wallis test, followed by a Dunn's multiple comparisons test. Bars represent the median and 95% confidence interval. **d** Velocities of WT or mutant F58D CRACR2a with 2 µM free Ca$^{2+}$, statistically analyzed using an unpaired $t$ test and two-tailed $p$ values were calculated. Data is pooled from three replicates (WT $n = 326$; F58D $n = 95$). Bars represent the median and 95% confidence interval; $p = 0.0579$. **e** Histogram of run lengths for WT and mutant F58D Halo-CRACR2a-positive particles with 2 µM free Ca$^{2+}$. The data were analyzed using a Mann–Whitney test and two-tailed $p$ values were calculated (****$p < 0.0001$; n.s. nonsignificant or $p > 0.05$). The data were pooled from three replicates (WT $n = 317$; F58D $n = 87$). Source data for panels **b**–**e** are provided in the Source Data file.

Fig. 6). Thus, on the LIC1 side, the interaction always involves the same amphipathic helix (human LIC1 residues 433–458), implying that the interactions of different adaptors are mutually exclusive. Four highly conserved amino acids (L444, F447, F448, and L451) on the hydrophobic face of the LIC1 helix insert into hydrophobic clefts in the adaptors (Fig. 6a–c). On the adaptor-side, however, the interaction not only involves different folds but also substantially different protein-protein contacts (Supplementary Fig. 6). Despite these structural differences, quantitative analysis of the interactions of seven adaptors, representing the three subfamilies, reveals a relatively narrow range of binding affinities for the LIC1 helix, with $K_D$s ranging from 1.5 to 15.0 µM (Figs. 1f and 2d)[18]. Note, however, that these

affinities are those of monomers. Because all the adaptors form coiled coil dimers, the actual affinities within the context of adaptor–dynein–dynactin complexes are likely enhanced through avidity.

While all the adaptors have the potential to bind two LIC1 helices, it is unclear how adaptors that recruit two dynein dimers use the dimeric character of the LIC1 interaction for activation. Indeed, there is no direct overlap between the crystal structures of adaptor–LIC1 helix complexes and the cryo-EM structures of adaptor–dynein–dynactin complexes[5,9]. Yet, the cryo-EM structure of BICDL1–dynein–dynactin was determined at higher resolution (3.5 Å), and reveals side chains for a portion of mouse BICDL1 (residues R132–N209), which partially overlaps with the

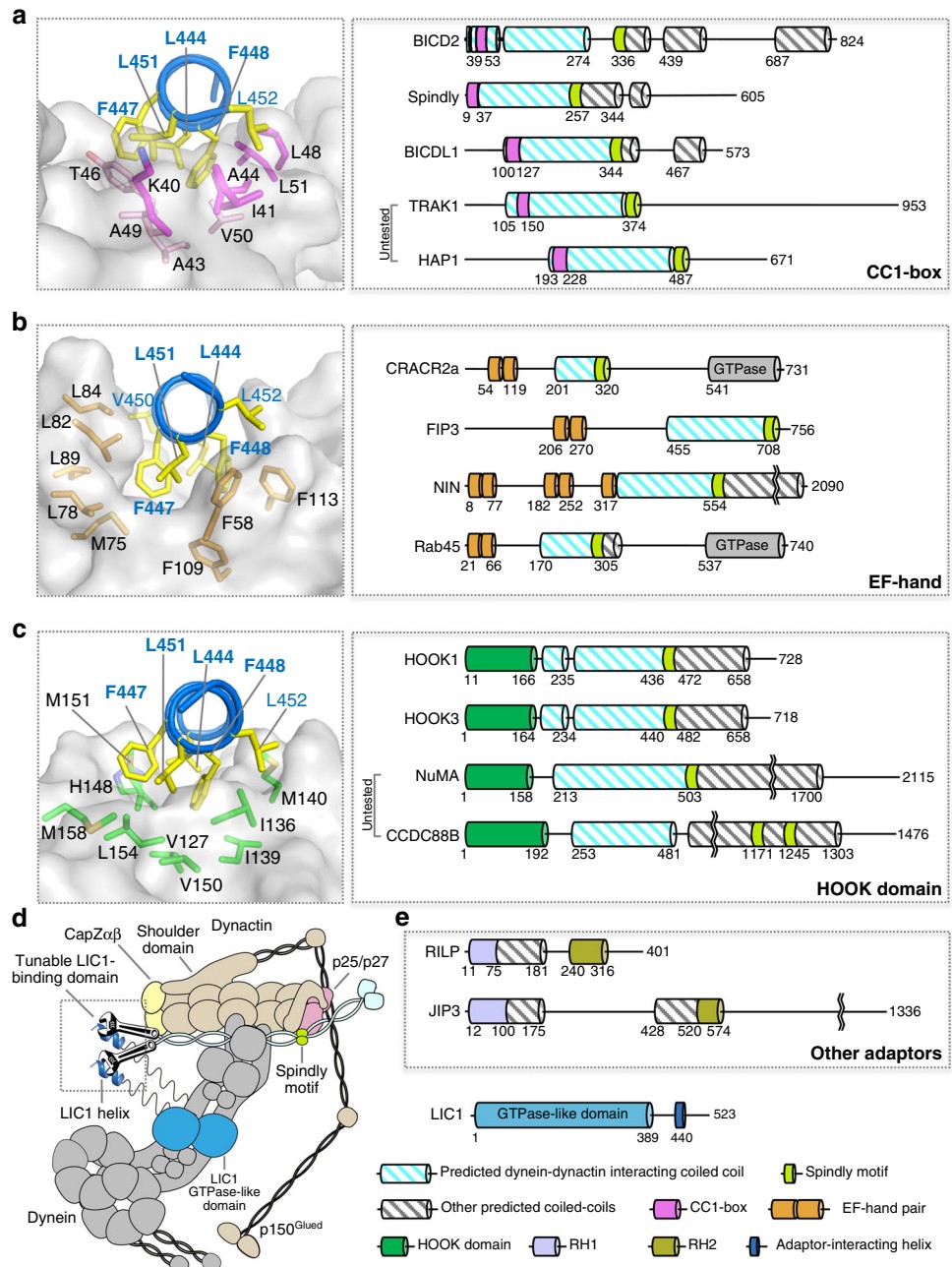

**Fig. 6 Classification of dynein–dynactin adaptors into three subfamilies with different LIC1 interactions. a–c** Residues L444, L451, F447, and F448 (bold) on the hydrophobic side of the LIC1 helix insert into a hydrophobic cleft in all the adaptors, which use different folds and different protein–protein contacts to binds this helix (see also Supplementary Fig. 6)**:** CC1-box (magenta), EF-hand pair (orange), and HOOK domain (dark green). Based on this interaction, the adaptors fall into three subfamilies, each containing several members with different domain architectures (for a complete list see Supplemental Table 1). C-terminal to the LIC1-binding site, all the adaptors present a coiled-coil segment (striped cyan) that binds the dynactin actin-like filament and provides part of the binding interface for the dynein tails[5,9,12]. C-terminal to the dynactin-binding coiled-coil, the adaptors have the Spindly motif (light green) that interacts with subunits p25/p27 at the pointed end of the dynactin complex (see also Supplementary Fig. 5). **d** Schematic representation of the LIC1–adaptor interaction within the context of dynein–dynactin–adaptor complexes. A prototypical adaptor is depicted (cyan), showing the variable (tunable) interaction with the LIC1 helix near CapZαβ at the barbed end of the dynactin filament, the coiled-coil segment that binds dynein–dynactin, and the Spindly motif. C-terminal to the Spindly motif the adaptors differ the most. This variable region is directly or indirectly responsible for the recruitment of specific cargoes. **e** Members of the RILP and JIP family share some features with the dynein–dynactin adaptors, including the ability to bind LIC1, but are unlikely to activate processive motility[1].

structure of BICD2 determined here. By aligning the sequences of human BICD2 and mouse BICDL1 for the region N-terminal to the Spindly motif, which share only 16% sequence identity, we were able to superimpose the crystal structure of BICD2-LIC1 onto the cryo-EM structure of BICDL1–dynein–dynactin (Supplementary Fig. 7). The superimposition shows that for CC1-box-containing adaptors the interaction with the LIC1 helix occurs at the level of CapZαβ. Given the distance between the binding site of the LIC1 helix and the Ras GTPase-like domains of the four LIC1 chains bound to the two dynein dimers, the most

likely scenario is that each dynein dimer contributes one LIC1 helix to the interaction. In this way, a single adaptor can simultaneously activate two dynein dimers. Because the separation between the HOOK domain and the Spindly motif is also ~270 amino acids, a very similar scenario can be proposed for most of the HOOK-domain containing adaptors (note that the location of the Spindly motif in CCDC88B in unclear) (Fig. 6c). The mechanism of activation for EF-hand-containing adaptors is less clear, since there is no apparent consistency in the location of the LIC1-binding EF-hand pair with respect to the Spindly motif among members of this subfamily (Fig. 6b).

Is the adaptor–LIC1 interaction regulated? Because the interaction with the LIC1 helix is primarily hydrophobic, regulation through phosphorylation of residues at the binding interface appears unlikely. Conformational changes in the adaptors appears more likely, and could be triggered by cargo binding at the C-terminus, as reported for BICD2[38], and analogous to the activation of several myosin and kinesin motors. Only CRACR2a is known to be activated by $Ca^{2+}$, and to promote processive dynein–dynactin motility in a $Ca^{2+}$-dependent manner[10]. We have shown here that CRACR2a is regulated by $Ca^{2+}$ binding to a single site within the EF-hand pair, which in turn triggers the binding of the LIC1 helix (Figs. 2–4). Disrupting this interaction through mutagenesis inhibits processive motility (Fig. 5). Combined these observations conclusively show that the LIC1–adaptor interaction is critical for activation of processive dynein motility, although not always necessary for the formation of a ternary complex with dynein–dynactin[10]. The $Ca^{2+}$-binding affinity of the site in CRACR2a ($K_D = 14.7\,\mu M$) is comparable to that of the low-affinity sites of the classical calcium sensors calmodulin and skeletal-muscle troponin C. These two proteins contain four $Ca^{2+}$-binding EF-hands, divided into two high-affinity sites with dissociation constants in the nanomolar range and two low-affinity sites with dissociation constants in the micromolar range[25,39,40]. The high-affinity sites are thought to be filled with $Ca^{2+}$ (or $Mg^{2+}$) under physiological conditions, whereas the low-affinity sites have high specificity for $Ca^{2+}$ and respond to fluctuations in $[Ca^{2+}]$ within the cellular milieu. The $Ca^{2+}$-binding site of CRACR2a seems to fall into the latter category, and thus may act as a regulatory site in cells. By analogy with troponin C[39], the $Ca^{2+}$-binding affinity of the CRACR2a site is expected to increase in the presence of a binding partner, i.e., the LIC1 helix. Our analysis further shows that two other EF-hand-containing adaptors, FIP3 and NIN, are not regulated by $Ca^{2+}$ and this is due to substitutions of key amino acids within the $Ca^{2+}$-binding loop (Fig. 3b). Of particular importance are the amino acids at positions 1 (or X) and 12 (or −Z) of the $Ca^{2+}$-binding loop, which in EF-hands that bind $Ca^{2+}$ are almost invariably aspartic acid and glutamic acid. In the rare occasions when the amino acid at position 12 is substituted by aspartic acid, the cation specificity of the loop can switch from $Ca^{2+}$ to $Mg^{2+}$[26]. Interestingly, the first EF-hand of FIP3 (FIP3_EFa, residues 202–237) is canonical, except for the presence of aspartic acid at position 12 (Fig. 3b), and thus $Mg^{2+}$ could possibly regulate FIP3 in cells.

Two other proteins, JIP3 and RILP, contain regions of coiled-coil structure, and RILP has been shown to bind the C-terminal region of LIC1[17] (Fig. 6e). Based on structural similarity, the N-terminal RH1 domain of these proteins may bind the LIC1 helix somewhat analogous to CC1-box-containing adaptors, which could possibly activate dynein. However, JIP3 and RILP seem to lack the Spindly motif and their coiled-coil regions are short, such that they are unlikely to function as bona fide dynein–dynactin adaptors. In summary, the structural and functional analysis of dynein–dynactin adaptors supports the notion that a tunable LIC1–adaptor interaction may modulate dynein's motility in a cargo-specific manner.

## Methods

**Proteins.** The cDNAs coding for mouse BICD2 (UniProt ID: Q921C5-1), human FIP3 (Uniprot: O75154-1), and human NIN (UniProt ID: Q8N4C6-1) were purchased from DNASU (Tempe, AZ). The cDNAs coding for human CRACR2a (UniProt ID: Q9BSW2-1) was a gift from Yousang Gwack (UCLA). Primers used in cloning are listed in Supplementary Table 2. Constructs $BICD2_{1-98}$, $CRACR2a_{47-122}$, $FIP3_{206-270}$, $NIN_{180-356}$, and $NIN_{1-87}$ were cloned into vector pMAL-c2X (NEB, Ipswich, MA). Human MBP-$LIC1_{FL}$ and MBP-$LIC1_{433-458}$ were obtained as described[18]. Heterodimeric (mutant/WT) $BICD2_{1-98}$ constructs were obtained by adding the heterodimerizing ZIP−/ZIP+ leucine zipper[24] using overlapping primers, while preserving the register of the coiled-coil heptad repeat. The fusion constructs were cloned into vectors pMAL-c2X and pET28a (NEB, Ipswich, MA), using the two different affinity tags (MBP- and His-tag) for equimolar purification of heterodimeric species.

The heterodimeric leucine zipper-BICD2 constructs were expressed in *Escherichia coli* BL21 (DE3) cells (Invitrogen, Carlsbad, CA) and grown in Terrific Broth medium at 37 °C until the $OD_{600}$ reached a value of 1.5–2, followed by 16 h at 19 °C in the presence of 0.25 mM isopropyl-β-D-thiogalactoside. The cells were collected by centrifugation, resuspended in 20 mM Tris (pH 7.0), 100 mM NaCl, 4 mM benzamidine hydrochloride, 1 mM PMSF, and 1 mM DTT and lysed using a Microfluidizer large-scale homogenizer (Microfluidics, Newton, MA). The proteins were purified through an amylose-affinity column according to the manufacturer's protocol (NEB). The MBP tag was removed by incubation with TEV protease overnight at 4 °C. The proteins were then loaded on a Ni-NTA-affinity column equilibrated with 50 mM Tris (pH 8.0), 500 mM NaCl, 4 mM benzamidine hydrochloride, 1 mM PMSF, and 1 mM TCEP (Buffer A). The proteins were eluted with Buffer A, supplemented with 300 mM Imidazole, and then additionally purified by gel filtration on a SD200HL 26/60 column in 20 mM Tris (pH 7.0), 100 mM NaCl, and 1 mM DTT. Point mutations were introduced using the QuikChange site-directed mutagenesis kit (Agilent Technologies, Wilmington, DE). All the other proteins were similarly expressed in *E. coli* BL21 (DE3) cells, followed by amylose-affinity purification and MBP-tag removal. For selenomethionine-substituted $BICD2_{1-98}$ and $CRACR2a_{47-122}$, cells were grown in M9 minimal media (Thermo Fisher Scientific, Waltham, MA), supplemented with selenomethionine (Molecular Dimensions, Maumee, OH), glucose, vitamins, and amino acids (except L-methionine). Proteins were then additionally purified by gel filtration on a SD200HL 26/60 column (GE Healthcare, Chicago, IL) in 20 mM Tris (pH 7.0), 100 mM NaCl, and 1 mM DTT. For $CRACR2a_{47-122}$ samples used in crystallization, 1 mM $CaCl_2$ was added to the buffer.

**Isothermal titration calorimetry.** ITC measurements were carried out on either a VP-ITC instrument (MicroCal, Northampton, MA) or an Affinity ITC instrument (TA Instruments, New Castle, DE). Protein samples were dialyzed for 3 d against 20 mM HEPES (pH 7.5), 100 mM NaCl, 0.5 mM TCEP (ITC buffer) and either 5 mM $CaCl_2$ or 5 mM EGTA. The $LIC1_{433-458}$ peptide was subjected to three cycles of solubilization/lyophilization in methanol 50% (v/v) to remove any trifluoroacetic acid and acetonitrile remaining after reverse-phase purification, followed by final solubilization in ITC buffer. The concentration of the peptide was determined by fluorescence with fluorescamine-labeled $LIC1_{433-458}$. The proteins (or the $LIC1_{433-458}$ peptide) in the syringe were titrated at a concentration 10 to 20 fold higher than that of the proteins in the ITC cell of total volume 1.44 ml (VP-ITC) or 0.94 ml (Affinity ITC), as indicated in the figures. Titrations consisted of 10 μl injections, lasting for 10 s, with an interval of 300–400 s between injections. The heat of binding was corrected for the heat of injection, determined by injecting proteins into buffer. Data were analyzed using the programs Origin (OriginLab, Northampton, MA) or Nanoanalyze (TA Instruments, New Castle, DE). The temperature and parameters of the fit (stoichiometry and affinity) of each experiment are given in the figures.

**Circular dichroism spectroscopy.** CD spectra were recorded from proteins and buffer only on a Chirascan-Plus CD Spectropolarimeter (Applied Photophysics, Leatherhead, UK) in 1 nm steps from 260 to 190 nm, using a 0.1 cm path-length quartz cuvette and a scan rate of 20 nm $min^{-1}$. The proteins ($NIN_{1-87}$ and $FIP3_{206-270}$) were concentrated to 100 μM in 20 mM Tris (pH 7.5), 100 mM NaCl, and 1 mM TCEP. Reported spectra correspond to merged spectra after buffer subtraction.

**Crystallization and structure determination.** $BICD2_{1-98}$ at ~10 mg $ml^{-1}$ in 10 mM Tris (pH 7.5), 100 mM NaCl, and 0.5 mM DTT was mixed with a 1.5 molar excess of $LIC1_{433-458}$ at 4 °C for 1 h. Crystals were obtained at 16 °C using the hanging-drop vapor diffusion method. The crystallization drop consisted of a 1:1 (v/v) mixture of protein solution and well solution containing 16% (v/v) iso-propanol, 80 mM sodium citrate tribasic dihydrate (pH 5.5), and 18% (w/v) polyethylene glycol 4000. $CRACR2a_{47-122}$ at ~10 mg $ml^{-1}$ in 10 mM Tris (pH 7.5),

100 mM NaCl, 0.5 mM DTT, and 1 mM $CaCl_2$ was mixed with a 1.5 molar excess of $LIC1_{433-458}$ at 4 °C for 1 h. Crystal were obtained at 16 °C using the hanging-drop vapor diffusion method. The crystallization drop consisted of a 1:1 (v/v) mixture of protein solution and well solution containing 16% (w/v) polyethylene glycol 10000, 0.5 M Bis–Tris (pH 6.5), and 10 mM $CoCl_2$. The quality of the crystals was further improved by microseeding. Crystals with selenomethionine-substituted $BICD2_{1-98}$ and $CRACR2a_{47-122}$ were obtained under identical conditions as those used for the WT proteins. Crystals of both complexes were flash-frozen in liquid nitrogen using a cryo-solution consisting of crystallization buffer supplemented with 30–35% (v/v) glycerol. X-ray datasets were collected from selenomethionine-substituted crystals at the Cornell High Energy Synchrotron Source beamline F1 ($BICD2_{1-98}$–$LIC1_{433-458}$ complex) and the Stanford Synchrotron Radiation Lightsource beamline 14-1 ($CRACR2a_{47-122}$–$LIC1_{433-458}$ complex). The diffraction datasets were processed using HKL2000[41]. Anisotropy data correction with HKL2000 eliminated weak, unreliable reflections, which improved the quality of data used in structure determination and refinement, but reduced completeness. The structures were determined ab initio using the single-wavelength anomalous dispersion method. The program SnB[42] was used to find the coordinates of the Se atoms in the structures. Model building was carried out with the program Coot[43]. The structures of BICD2–LIC1 and CRACR2a–LIC1 were refined with the programs Phenix[44] and Refmac5[45], respectively. Data collection and refinement statistics are listed in Table 1. Figures were generated with the program PyMOL (Schrödinger, New York City, NY). Sequence alignments were generated with the program MAFFT[46], and illustrated using either Jalview[47] or ESPript[48]. The conservation scores of amino acids in sequence alignments were determined with the program Scorecons[49].

**Multi-angle light scattering**. Samples were separated by size exclusion chromatography on a Superose 6 10/300 GL column (GE Healthcare) equilibrated with 20 mM Tris (pH 7.5), 100 mM NaCl, and 1 mM DTT, using an Agilent 1100 HPLC system (Agilent Technologies, Santa Clara, CA). Light scattering was measured in-line, using a DAWN-HELEOS multi-angle light scattering detector and an Optilab rEX refractive index detector. The scattering data were analyzed with the program ASTRA (Wyatt Technology, Santa Barbara, CA).

**Single-molecule motility assays**. The movement of CRACR2a-containing complexes from cell extracts was tracked using TIRF microscopy[7]. Motility assays were performed in flow chambers constructed with a glass slide and a coverslip silanized with PlusOne Repel-Silane ES (GE Healthcare), held together with vacuum grease to form a ~10 μl chamber. Rigor kinesin-$1_{E236A}$ (0.5 μM) was non-specifically absorbed to the coverslip[50] and the chamber was then blocked with 5% pluronic F-127 (Sigma-Aldrich). GMPCPP microtubule seeds (250 nM), labeled at a 1:40 ratio with HiLyte Fluor 647 (Cytoskeleton, Denver, CO), were flowed into the chamber and immobilized by interaction with rigor kinesin-$1_{E236A}$. Free tubulin (11.25 μM), labeled at a 1:20 ratio with HiLyte Fluor 488 tubulin, was added with the lysate to grow dynamic microtubules from the seeds. HeLa cells grown in 10 cm plates to 70–80% confluence and expressing full-length Halo-tagged CRACR2a WT, CRACR2a mutant F58D, or HaloTag alone were labeled with TMR-HaloTag ligand (Promega, Madison, WI) 18–24 h post transfection (Supplementary Fig. 4b). Cells were then lysed in 100 μl lysis buffer [40 mM HEPES (pH 7.4), 120 mM NaCl, 1 mM EDTA, 1 mM ATP, 0.1% Triton X-100, 1 mM PMSF, 0.01 mg ml$^{-1}$ TAME, 0.01 mg ml$^{-1}$ leupeptin, and 1 μg ml$^{-1}$ pepstatin-A]. Cell lysates were clarified by centrifugation (17,000$g$) and diluted in P12 motility buffer [12 mM PIPES (pH 6.8), 1 mM EGTA, and 2 mM $MgCl_2$] supplemented with 1 mM ATP, 1 mM GTP, 0.08 mg ml$^{-1}$ casein, 0.08 mg ml$^{-1}$ bovine serum albumin, 2.55 mM DTT, 0.05% methylcellulose, and an oxygen scavenging system (0.5 mg ml$^{-1}$ glucose oxidase, 470 U ml$^{-1}$ catalase, and 3.8 mg ml$^{-1}$ glucose). Cell lysates were then incubated for 5 min with either 2 μM free $Ca^{2+}$ or 2 mM EGTA before flowing into the imaging chamber to examine the dynein–dynactin driven motility of single Halo-CRACR2a-positive particles (WT and mutant F58D CRACR2a) in the absence or the presence of $Ca^{2+}$. We used the program Maxchelator[51] to calculate the concentration of $CaCl_2$ needed to achieve 2 μM free $Ca^{2+}$ in solution. All the videos (2 min, 4 frames s$^{-1}$) were acquired at 37 °C using a Nikon TIRF microscopy system (Perkin Elmer, Waltham, MA) on an inverted Ti microscope equipped with a 100× objective and an ImageEM C9100-13 camera (Hamamatsu Photonics, Hamamatsu, Japan) with a pixel size of 0.158 μm and controlled with the program Volocity (Improvision, Coventry, England). At least 5 microtubules per video were analyzed by generating kymographs using the Multi Kymograph plugin of ImageJ2 (https://imagej.net/ImageJ2) and analyzed in Excel (Microsoft, Redmond, WA). During data acquisition, the seeds (HiLyte 647-labeled) were imaged at a rate of 1 frame min$^{-1}$ and free tubulin (HiLyte Fluor 488-labeled) at 12 frames min$^{-1}$. Only non-bundled microtubules that had one end growing from the seed clearly faster (+ end) than the other (− end) were considered, and only movements toward the slow growing end of these microtubules were analyzed. At least 15 microtubules were analyzed per replicate and three biological and technical replicates were performed for a final $n = 123$ microtubules per condition. Runs ending within 0.5 μm of the microtubule minus end were excluded from the run length analysis. Statistical analyses were performed in Prism (GraphPad, San Diego, CA). A two-tailed $t$ test was used for velocity analysis, a two-tailed Mann–Whitney

test was used for run-length analysis, and a Kruskal–Wallis test followed by a Dunn's multiple comparisons test was used for analysis of the number of events.

**Reporting summary**. Further information on experimental design is available in the Nature Research Reporting Summary linked to this paper.

## Data availability

Atomic coordinates and structure factor amplitudes for the structures of BICD2-LIC1 and CRACR2a-LIC1 were deposited with the Protein Data Bank (PDB) under accession codes 6PSE and 6PSD, respectively. Other data and materials are available from the corresponding author upon reasonable request. Source data are provided with this paper.

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

## Acknowledgements
This work was supported by National Institutes of Health (NIH) grants RM1 GM136511 to R.D. and R35 GM126950 to E.L.F.H. and National Science Foundation Graduate Research Fellowship DGE-1845298 to S.E.C. Data collection at the Stanford Synchrotron Radiation Lightsource (SSRL) beamline 14-1 was supported by U.S. Department of Energy contract DE-AC02-76SF00515 and NIH grant P41 GM103393. Data collection at the Macromolecular x-ray science at the Cornell High Energy Synchrotron Source (MacCHESS) beamline F1 was supported by NSF grant DMR-1332208 and NIH grant GM-103485. The authors thank Zenon Grabarek for helpful feedback about EF-hand proteins.

## Author contributions
I.-G.L. and R.D. designed and performed the biochemical and structural experiments. E.L.F.H. and S.E.C. designed and performed the in vitro motility experiments. S.S.A. participated in the protein preparation. I.-G.L. and R.D. wrote the paper. All the authors reviewed the figures and paper and approved its final version.

## Competing interests
The authors declare no competing interests.
