## [Peer Review File · Nature Communications]

REVIEWER COMMENTS

Reviewer #1 (Remarks to the Author):

The authors of this manuscript set out to understand the structural basis for dynein Light Intermediate Chain (LIC) binding to the dynein activating adaptors BicD2 and CRACR2a. This work is a nice complement to a previous publication from the groups investigating the structural basis for Hook3-LIC binding (Lee, et al, 2018). In line with their previous publication, the authors confirm that the small stretch of helix in LIC drives binding to BicD2 and CRACR2a. The authors confirm a binding stoichiometry of 2 LICs to 1 dimer of activating adaptor using ITC, which again, supports their findings in Lee, et al. They probe whether Ca²⁺ affects binding of LIC to other activators that have EF hands (NIN, FIP3, and CRACR2a) and find that only CRACR2a binding is regulated by modulating calcium levels. Finally, by solving the structure of LIC1-aa 433-458 bound to truncated versions of BicD2 and CRACR2a, they show that the small helix in LIC is able to recognize vastly different structures of activating adaptors.

This work provides an important contribution to the dynein biology field. While a lot of the results of many of the experiments (in light of their previous publication) are not surprising, the crystal structures of two additional activating adaptors bound to LIC are important and add significantly to our understanding of how LIC binding to activating adaptors may promote dynein activity. Pending the points below, I recommend this manuscript for publication.

Major Comments:

1. I find the result that FIP3 and NIN are not Ca²⁺ binding proteins and thus association with LIC is not regulated by Ca²⁺ to be very interesting, especially when contrasted with the result that CRACR2a is regulated by Ca²⁺. My concern however is that there is no assay used to ensure that the FIP3 and NIN peptides are folded properly. This is especially important since the result observed is a negative one. It is unlikely that these peptides are completely misfolded (as they both bind LIC). However, it is possible that the truncated NIN and FIP3 do not fully assume a structure that can bind to Ca²⁺. While not a perfect experiment, circular dichroism of NIN, FIP3, and CRACR2a would confirm that the peptides are all mostly helical, as would be expected for an EF hand.

A more compelling experiment (though outside of the reasonable expectation for acceptance) would be to transiently transfect full length NIN or FIP3 (or longer truncations), purify from human cell culture lysates, and use a MS/MS to identify if Calcium metal co-purifies. If it does not, this would convince me that calcium plays no role in regulating NIN or FIP3 activity.

2. Along this note, do the authors predict that LIC can bind any protein with an EF-hand? How conserved are the amino acids that make up the hydrophobic path in CRACR2a that bind to LIC? Authors should include a sequence alignment of these residues for many EF hand-containing proteins. If they are highly conserved, authors should test if LIC retains the ability to bind any EF-hand containing protein (using the ITC binding experiment that has been used throughout the paper).

3. The least compelling part of the manuscript is figure 5. There are a number of items that need to be addressed within this figure.

a. First, the motility seen in the representative kymographs does not look very convincing. Only two (very, very short) processive runs are seen in the kymographs and these look saltatory- not like the normal long fast runs that activated dynein has shown in other publications. Better (and less cropped) representative kymographs should be shown.

b. The authors should include a negative control (just the labeled Halo tag, eg) for their pull-down

motility experiments to ensure that the motor they are pulling down is specific to CRACR2a and not just non-specifically co-eluting with beads. They should also include a positive control, like Hook3 or BicD2, which are better characterized than CRACR2a and the velocity of dynein activated by Hook3 or BicD2 should not be affected by changing calcium levels in the buffer.

c. The authors should ensure that the motor they are pulling down is dynein (with a western blot of their IP). If this is not possible, the authors should use polarity marked microtubules to ensure that the runs they observe are moving toward the minus-end

d. Have they blotted for kinesins to rule out that CRACR2a is pulling down opposite polarity motors? If the authors do a polarity marked microtubule experiment and observe only minus-end directed runs, than this is not necessary. If they do not do that experiment, then they should either do mass spec or blot for a number of kinesin motors (especially Kif5s) to ensure that observed motility is not kinesin driven. In their western in Sup Figure 4 b, their blot contains a band for Kif5b, but I believe that this is just a halo-tagged construct pulled out of another cell lysate that they are using for size reference.

e. The statistics in the 5a graph are not correct. The authors do a paired t test, which is not appropriate for these experiments whose outcomes are independent. They also make multiple comparisons between conditions (ie compare WT with CaCl₂ to F58D with CaCl₂, then compare WT with CaCl₂, to WT without calcium). To do multiple comparisons between experiments, authors cannot just do a t-test. I believe that the correct statistical test is an ANOVA or an unpaired t test with a Bonferroni correction.

f. In Figure 5a graph, what does each dot refer to? Are these separate technical replicates? How many bioreplicates were performed? This should be clearly stated either in the legend or the methods. At least two bioreplicates (completely separate IPs) should be performed per construct.

g. The total number of events observed in Figure 5a is very low. The Y-axis is number of events per 100uM. For their most active construct (wt with calcium) the most number of events observed per tech rep is 2 events per 100um. That is very sparse. Why are they observing such low numbers of moving events? Given how sticky dynein can be, it worries me that robust motility isn't observed for their most active sample. Here is where comparison to a positive control and an included negative control would lead credence to their data.

h. The authors should show that the F58D mutation actually prevents binding between CRACR2a and LIC using the ITC assay they have used previously.

i. An analysis of velocity of the runs that were observed with the F58D construct with Calcium present should be performed. This could be added to Figure 5b.

4. Why do the authors think that different activating adaptors have different LIC binding domains? Is this an accident of evolution or do the authors speculate that the different types of LIC binding domains confer some additionally regulatory abilities. In the case of CRACR2a, it is clear how calcium can be harnessed to regulate dynein activation, but for the other activators, it's not clear. It would be great if the authors could speculate a little more about this in the discussion.

Minor comments:

1. "Adaptor" should be changed to "activating adaptor" in all cases. For non-dynein experts reading this study, adaptor may be a confusingly general term.

2.Space allowing, it may be nice to move the ZIP+/ZIP- cartoon to the main figure. It really clarifies the experimental design.

Reviewer #2 (Remarks to the Author):

Cytoplasmic dynein-1 is able to move processively on microtubules by virtue of its interaction with dynactin and a coiled-coil adaptor. Multiple protein-protein interactions within this complex are required for motility, including between the dynein light intermediate chain (LIC) and the adaptor. This specific interaction is unique since there is great variability between adaptor proteins where they bind the LIC. In this manuscript, Lee et al. solve the crystal structure of representatives from two of the three largest subfamilies of adaptor proteins, as categorized by their LIC interaction domains, showing how they bind the LIC at the atomic level. Isothermal titration calorimetry (ITC) experiments of members of the EF-hand-containing subfamily further demonstrate how calcium can regulate LIC binding. By combining these data with previous insights from the same group (Lee et al., 2018), the authors provide a complete rulebook for the adaptor-LIC interaction. These data provide strong evidence for a regulatory role for this interaction in cargo recruitment and transport. The data are of high quality and I recommend publication of the manuscript, with some suggestions:

Major Comments

1. The authors show that a mutation in CRACR2a (F58D) results in a reduced numbers of motile dynein complexes, but do not show that this is due specifically to a disruption of its interaction with the LIC. The authors should confirm this with ITC, as was done with BICD2 (Y46D).
2. The authors have compiled an impressive library of LIC1 interactions and binding affinities. Can the authors extract consensus interaction elements that are common to the LIC-binding domains? In line with the spirit of the manuscript, it should be possible to discuss how the conformation and/or identify of the LIC-binding residues correlate with binding affinities. This would show definitively that the interaction is "tunable", and could even be used as a tool for discovering new adaptors.
3. The in vitro motility data analysis should be clarified. Assessing the number of motile events per 100 um suggests that all experiments were performed for the same amount of time, but the length of the movies are not stated. It's also not clear if the data points in Figure 5a are from different lysates or from individual microtubules. Clarifications in the legend and methods would suffice.

Minor Comments

1. In Figure 4a, it might be helpful to include "EFb" somewhere to remind the reader that this structure is different from EFa.
2. In Figure 5a, it might be helpful to include a definition of "MT".
3. In Figure 5b, the y-axis label is "Frequency" but the data is rather a probability. Furthermore, the legend should describe if the data are pooled from different experiments. As another small point here, the authors should consider whether the experiment has the precision of three significant digits, as shown in the mean velocity.
4. The methods for how the "number of motile events" was calculated refers to this as a "landing rate" (line 495), which can be interpreted as including non-motile events. This should be clarified.
5. In lines 389-390, it's not clear what "as described above" is referring to, given that this is the first mention of protein expression.
6. In Supplementary Figure 1b, it's not clear why some e and g positions are not highlighted; were they intentionally left out?
7. The authors should state the pixel size of their single molecule data, without which one cannot assess the validity of the 0.5 um criterion for classifying motile events.
8. There is an extra comma on line 390 ("After, ...").

Reviewer #3 (Remarks to the Author):

Lee et al. investigate how cargo adaptor proteins activate the motility of dynein, a microtubule-based molecular motor, through their interactions with one of the dynein subunits, the light intermediate chain 1 (LIC1). Dynein transports a large diversity of cargos and this motor does so by binding to many different adaptor proteins, each bound to a specific cargo. The authors focus on the adaptor-dynein LIC1 interaction to understand what commonalities are shared among all the cargo adaptors, which have little sequence homology, and how they may differ to “tune” dynein’s transport for specific cellular contexts. By taking a comparative structural approach and using a combination of crystallography, binding assays, and in vitro motility assays, they find that two adaptors exhibit a hydrophobic-binding cleft into which a conserved C-terminal amphipathic helix in LIC1 inserts. The authors also delve into how the adaptor-LIC1 interaction is “tunable” and largely focus on two factors: affinities and Ca²⁺ regulation. This study expands upon their previous work published in Nature Communications, which identified the LIC1 helix that binds several cargo adaptors. Overall, this paper effectively expands upon and integrates the information we have thus far on the adaptor-dynein interaction and further demonstrates LIC1’s importance in dynein motility. I think the paper will be of interest to those studying dynein and cargo transport, as well as the broad Nature Communications readership. However, I recommend that a few issues be addressed before publication.

Major comments:

1. ITC experiments:

The authors focus on the Ca²⁺ binding of CRACR2a, yet in these binding assays, CRACR2a is tagged with MBP. Although a tag is necessary for the detection of CRACR2a (via 280 nm-absorbance), an MBP tag alone is lacking as a negative control. MBP has been shown to bind Ca²⁺ (see Behbehani et al. 2008). Additional data in this study, particularly the structure of CRACR2a co-crystallized with Ca²⁺, supports that Ca²⁺ does directly bind CRACR2a, however, to obtain an accurate affinity for Ca²⁺, the authors need to incorporate MBP alone in their ITC assay (Fig. 3).

For the ITC experiment, LIC1 (aa 433-458) is purified by reverse phase chromatography in acetonitrile; thus, presumably the LIC1 helix is largely denatured and the affinities measured in this assay may change upon the use of folded protein. Since the varying affinities of cargo adaptors is a large point of discussion in the paper to explain how adaptors tune dynein motility, this caveat should be noted in the text.

2. Single-molecule motility assays:

The authors’ motility data expands upon the study Wang et al. 2019, which showed CRACR2a activates dynein motility and is regulated by Ca²⁺; the authors here pinpoint LIC1 as a major contributing factor. However, the wild-type (WT) control used by Lee et al. does not appear to have comparable motility to the WT shown in Wang et al. There are few events of processive motility in the WT control (Fig. 5A). Are the kymographs truly representative of WT motility? There is one notable difference between the two studies: Wang et al. used purified protein, whereas here diluted cell lysate is used and the effective protein concentration in the assay is unclear. Although the authors successfully demonstrate a difference between WT and the F58D CRACR2a mutant, there is little dynamic range in which to explore motility differences since WT already has so few motile events. If WT motility cannot be improved upon by the authors, then the differences between the assays in the two studies should be noted in the text. Also, the y-axis of the graph in A needs to be in “per min per μm microtubule” for easier interpretation.

This in vitro data would be nicely complemented with a demonstration of the expected weakened interaction between LIC1 and the CRACR2a F58D mutant—either by ITC or a pulldown—to further support the chosen structure-inspired mutation.

Minor comments:

1. Table 1: Given the BICD2 dataset is anisotropic, the authors should add a few more statistics to

be clear about the quality of their dataset. The authors should report the effective resolution (deff, see Weiss, M. S. Global indicators of X-ray data quality. *J Appl Crystallogr*, 2001) and the resolution estimates along the three axes, which can be obtained from the anisotropy servers StarAniso or Xtrige. Finally, the redundancy and CC1/2 for both crystal structure datasets should be added to Table 1 to further indicate data quality.

2. Lines 102-107: If the structure is reminiscent of the RILP homology-1 domain, then the authors should show this comparison. Otherwise, this seems speculative and tangential.

3. Fig. 1C: I do not think this is the best way to illustrate the point that A43 and G47 are unusually small and create a cavity. A surface model representation of the binding interface would be more appropriate to show an actual cavity. Alternatively, move the alignment in Supplemental Fig. 1A, which shows the strong conservation of A43 and G47, to the main figure.

4. Fig. 1E: The interactions should be labeled with the distances to illustrate why Y46 was selected for further study.

5. Fig. 3B: The NIN alignment shows lack of Ca²⁺-binding residues, but FIP3 EFa does not despite being denoted as "non-Ca²⁺-binding" (this is actually noted in the discussion—line 369). Thus, I'm not sure if this section of the figure truly adds anything, certainly not for a main figure and could be moved to the supplement. Even if you look at the Ca²⁺-binding residues in the true Ca²⁺-binding EF hands, not all of these residues highlighted as highly conserved are actually conserved, making it feasible that the FIP3 EFa domain may bind Ca²⁺ (perhaps upon binding an accessory protein). Note also, that one of the Ca²⁺-binding aspartic acids is conserved in FIP3 EFa but not highlighted purple to indicate conservation.

6. Fig. 4D: I do not think it is necessary to show both troponin and calmodulin in the figure to show CRACR2a is binding Ca²⁺ similarly; perhaps only one should be included. If there is good reason for showing troponin and calmodulin, the authors should state this clearly in the text.

7. Line 249: The authors should state the rationale for why F58 was mutated to specifically aspartic acid and not a different amino acid, such as alanine.

8. Many figure legends are lacking the number of times the experiment was conducted. Authors should include "n =" to indicate reproducibility.

9. The discussion can be improved. Paragraphs 2-3 should be condensed because as written, it comes across as a review and is not directly put in the context of the current study. Also, a number of broad implications can be derived from this study, yet the last paragraph does not do this justice. The discussion should end on the wider implications at hand, rather than a focus on proteins that lack data to show they are true dynein-dynactin adaptors.

10. Fig. 6 and Supplemental table 1 are great additions to the study and will be useful for the dynein community.

REVIEWER COMMENTS

We thank the reviewers for their careful reading of the manuscript and suggestions that improve the paper. Several new experiments were added during revisions, which combined with restricted access to our research facilities during the lockdown explains the relatively slow turnaround time. Please, also note that the first author (Dr. In-Gyun Lee) took a faculty job in Korea, such that the newly-added ITC experiments were collected on a different instrument in Korea (as described in the Methods). Major changes in the manuscript are highlighted in yellow. Two new references were added in response to reviewer's comments. In the point-by-point response below, we use black text for the reviewers' comments and blue text for the authors' response.

Reviewer #1

The authors of this manuscript set out to understand the structural basis for dynein Light Intermediate Chain (LIC) binding to the dynein activating adaptors BicD2 and CRACR2a. This work is a nice complement to a previous publication from the groups investigating the structural basis for Hook3-LIC binding (Lee, et al, 2018). In line with their previous publication, the authors confirm that the small stretch of helix in LIC drives binding to BicD2 and CRACR2a. The authors confirm a binding stoichiometry of 2 LICs to 1 dimer of activating adaptor using ITC, which again, supports their findings in Lee, et al. They probe whether Ca²⁺ affects binding of LIC to other activators that have EF hands (NIN, FIP3, and CRACR2a) and find that only CRACR2a binding is regulated by modulating calcium levels. Finally, by solving the structure of LIC1-aa 433-458 bound to truncated versions of BicD2 and CRACR2a, they show that the small helix in LIC is able to recognize vastly different structures of activating adaptors.

This work provides an important contribution to the dynein biology field. While a lot of the results of many of the experiments (in light of their previous publication) are not surprising, the crystal structures of two additional activating adaptors bound to LIC are important and add significantly to our understanding of how LIC binding to activating adaptors may promote dynein activity. Pending the points below, I recommend this manuscript for publication.

Major Comments:

1. I find the result that FIP3 and NIN are not Ca²⁺ binding proteins and thus association with LIC is not regulated by Ca²⁺ to be very interesting, especially when contrasted with the result that CRACR2a is regulated by Ca²⁺. My concern however is that there is no assay used to ensure that the FIP3 and NIN peptides are folded properly. This is especially important since the result observed is a negative one. It is unlikely that these peptides are completely misfolded (as they both bind LIC). However, it is possible that the truncated NIN and FIP3 do not fully assume a structure that can bind to Ca²⁺. While not a perfect experiment, circular dichroism of NIN, FIP3, and CRACR2a would confirm that the peptides are all mostly helical, as would be expected for an EF hand. A more compelling experiment (though outside of the reasonable expectation for acceptance) would be to transiently transfect full length NIN or FIP3 (or longer truncations), purify from human cell culture lysates, and use a MS/MS to identify if Calcium metal co-purifies. If it does not, this would convince me that calcium plays no role in regulating NIN or FIP3 activity.

We performed the suggested circular dichroism (CD) experiments (**Supplementary Fig. 2b**). The CD spectra of FIP3₂₀₆₋₂₇₀ and NIN₁₋₈₇ are characteristic of α -helical structures, with minima

around 222 and 208 nm and no evidence of unfolding (lines 174-176). The two proteins also bind LIC1 with similar affinities as CRACR2a by ITC (**Fig. 2b-d**). Note that unfolded proteins behave in unmissable bizarre ways by ITC and are typically insoluble at the high concentrations used in these experiments. Thus, the evidence appears quite strong to rule out questions about folding. That FIP3₂₀₆₋₂₇₀ and NIN₁₋₈₇ do not bind Ca²⁺ is consistent with the lack of key Ca²⁺-binding residues in the EF-hands (**Fig. 3a,b**). This point is now further explained in the main text (lines 233-239).

2. Along this note, do the authors predict that LIC can bind any protein with an EF-hand? How conserved are the amino acids that make up the hydrophobic path in CRACR2a that bind to LIC? Authors should include a sequence alignment of these residues for many EF hand-containing proteins. If they are highly conserved, authors should test if LIC retains the ability to bind any EF-hand containing protein (using the ITC binding experiment that has been used throughout the paper).

We certainly do not predict that just any EF-hand protein would bind the LIC1 helix. The EF-hand domain displays great plasticity but also has exquisite specificity, with Calmodulin as a good example (Chin & Means, *Trends Cell Biol.* 2000). We actually show in the manuscript an example of a three EF-hand-construct (NIN₁₈₀₋₃₅₆) that does not bind the LIC1 helix (**Supplementary Fig. 2c**).

We performed the requested sequence alignment of EF-hand pairs that either bind or are shown or presumed not to bind the LIC1 helix (**Supplementary Fig. 3c**). This is a good addition and we thank the reviewer for the suggestion (lines 252-255). However, we cannot identify any specific conservation pattern among the residues found in the vicinity of the LIC1 helix in the structure of the complex with CRACR2a that would allow us to predict whether another EF-hand pair can bind the LIC1 helix. We do not believe it is reasonable to begin expressing just any EF-hand pair in large amounts and with high purity (as required for ITC), when there is no reason to suspect their role in the biological process under investigation, particularly that these proteins lack other features that characterize the dynein-dynactin adaptors.

3. The least compelling part of the manuscript is figure 5. There are a number of items that need to be addressed within this figure.

a. First, the motility seen in the representative kymographs does not look very convincing. Only two (very, very short) processive runs are seen in the kymographs and these look saltatory- not like the normal long fast runs that activated dynein has shown in other publications. Better (and less cropped) representative kymographs should be shown.

In agreement with these comments, we made several changes allowing us to improve the *in vitro* motility assays, including changes to the buffer (see Methods) and increased temperature (37 °C) and CaCl₂ concentration (2 μM free Ca²⁺ instead of 2 μM total Ca²⁺). In the new experiments, we also used dynamic microtubules, which grow longer than Taxol-stabilized microtubules and allow for polarity identification. The data now presented in **Fig. 5** is all new, including improved statistical analysis based on a greater number of longer runs (please, see also the response to point 3.e below). The corresponding sections in the Results and Methods have changed substantially.

b. The authors should include a negative control (just the labeled Halo tag, e.g.) for their pull-down motility experiments to ensure that the motor they are pulling down is specific to CRACR2a and not just non-specifically co-eluting with beads. They should also include a

positive control, like Hook3 or BicD2, which are better characterized than CRACR2a and the velocity of dynein activated by Hook3 or BicD2 should not be affected by changing calcium levels in the buffer.

Clearly, the Western blot we had presented in **Supplementary Fig. 4** was confusing. We did not perform pull-downs. Our lysate-based motility assays involve flowing the clarified cell lysate into the TIRF chamber (except most membranes, which are removed by centrifugation). Halo-tag alone is shown as a negative control (**Supplementary Fig. 4d**). BICD2 is routinely used in lab assays as a positive control; these data were not included since they have been published multiple times previously (see for example Lee et al., 2018; Olenick et al., 2019). Please, see response to point 3.c. below for how we ensure only dynein runs are measured.

c. The authors should ensure that the motor they are pulling down is dynein (with a western blot of their IP). If this is not possible, the authors should use polarity marked microtubules to ensure that the runs they observe are moving toward the minus-end

This is a good point, and to address it we repeated the entire set of TIRF experiments using dynamic two-color-labeled microtubules (HiLyte Fluor 488-labeled microtubules growing from HiLyte Fluor 647-labeled seeds), allowing us to clearly define the polarity of microtubules (**Supplementary Fig. 4c**). We see almost exclusively minus end-directed motility of Halo-tagged CRACR2a particles (95%). Only unambiguous minus-end directed runs were analyzed.

d. Have they blotted for kinesins to rule out that CRACR2a is pulling down opposite polarity motors? If the authors do a polarity marked microtubule experiment and observe only minus-end directed runs, than this is not necessary. If they do not do that experiment, then they should either do mass spec or blot for a number of kinesin motors (especially Kif5s) to ensure that observed motility is not kinesin driven. In their western in Sup Figure 4 b, their blot contains a band for Kif5b, but I believe that this is just a halo-tagged construct pulled out of another cell lysate that they are using for size reference.

Please, see the response to point 3.c. above. Using polarity-marked microtubules we found that 95% of the runs were minus end-directed. We also performed additional Western blots to confirm Halo-tagged CRACR2a expression (**Supplementary Fig. 4**).

e. The statistics in the 5a graph are not correct. The authors do a paired t test, which is not appropriate for these experiments whose outcomes are independent. They also make multiple comparisons between conditions (i.e. compare WT with CaCl₂ to F58D with CaCl₂, then compare WT with CaCl₂, to WT without calcium). To do multiple comparisons between experiments, authors cannot just do a t-test. I believe that the correct statistical test is an ANOVA or an unpaired t test with a Bonferroni correction.

In agreement with this comment, we now analyze the data using the Kruskal-Wallis test (a non-parametric test equivalent to ANOVA, since the data is right skewed) followed by a Dunn's multiple comparisons test of the data.

f. In Figure 5a graph, what does each dot refer to? Are these separate technical replicates? How many bioreplicates were performed? This should be clearly stated either in the legend or the methods. At least two bioreplicates (completely separate IPs) should be performed per construct.

Each data point in the new **Fig. 5c** represents the average motility observed along a single microtubule, with at least 15 microtubules analyzed per biological and technical replicate. We have performed 3 biological and technical replicates for a final n = 123 microtubules per condition.

g. The total number of events observed in Figure 5a is very low. The Y-axis is number of events per 100uM. For their most active construct (wt with calcium) the most number of events observed per tech rep is 2 events per 100um. That is very sparse. Why are they observing such low numbers of moving events? Given how sticky dynein can be, it worries me that robust motility isn't observed for their most active sample. Here is where comparison to a positive control and an included negative control would lead credence to their data.

As noted above, we agreed with the concerns raised about our initial data. We replaced the initial data with an entirely new dataset that exhibits more robust motility due to improvements in the assay conditions, including a more accurate calcium concentration and polarity-marked microtubules, as detailed in the Methods section.

h. The authors should show that the F58D mutation actually prevents binding between CRACR2a and LIC using the ITC assay they have used previously.

We expressed the F58D mutant within the context of construct MBP-CRACR2a (used in ITC experiments). We then confirmed by ITC that this mutant, used in motility assays, does not bind the LIC1 helix in the presence of Ca^{2+} (**Fig. 5a**). Motivated by a question from another reviewer, we also show that MBP does not bind Ca^{2+} (**Supplementary Fig. 2a**).

i. An analysis of velocity of the runs that were observed with the F58D construct with Calcium present should be performed. This could be added to Figure 5b.

We now include an analysis of both velocities and run lengths with WT and F58D in the presence of Ca^{2+} (**Fig. 5d,e**).

4. Why do the authors think that different activating adaptors have different LIC binding domains? Is this an accident of evolution or do the authors speculate that the different types of LIC binding domains confer some additionally regulatory abilities? In the case of CRACR2a, it is clear how calcium can be harnessed to regulate dynein activation, but for the other activators, it's not clear. It would be great if the authors could speculate a little more about this in the discussion.

This is an interesting question to which we do not have a firm response. Yet, the leitmotif of the entire paper is the proposal that this interaction indeed constitutes a regulatory element, an idea that is specifically developed in the Discussion, **Fig. 6** and **Supplementary Table 1**. We do not think this is *an accident of evolution*.

Minor comments:

1. "Adaptor" should be changed to "activating adaptor" in all cases. For non-dynein experts reading this study, adaptor may be a confusingly general term.

We find that in many cases the use of "activating adaptor" would be convoluted. For instance, "activating adaptor-dynein-dynactin complex". To avoid confusion, we have added a clarification in the Introduction: "*activating adaptors (hereafter referred to as adaptors)*", line 37.

2.Space allowing, it may be nice to move the ZIP+/ZIP- cartoon to the main figure. It really clarifies the experimental design.

We agree that the ZIP+/ZIP- cartoon is useful to clarify the experimental design, but we believe it belongs in Supplementary Information. **Fig. 1** is already quite busy, and we cannot add another figure to the paper for this cartoon, particularly that the ZIP+/ZIP- heterodimer design is published and it is not our work (Moll et al., *Protein Science* 2001).

Reviewer #2

Cytoplasmic dynein-1 is able to move processively on microtubules by virtue of its interaction with dynactin and a coiled-coil adaptor. Multiple protein-protein interactions within this complex are required for motility, including between the dynein light intermediate chain (LIC) and the adaptor. This specific interaction is unique since there is great variability between adaptor proteins where they bind the LIC. In this manuscript, Lee et al. solve the crystal structure of representatives from two of the three largest subfamilies of adaptor proteins, as categorized by their LIC interaction domains, showing how they bind the LIC at the atomic level. Isothermal titration calorimetry (ITC) experiments of members of the EF-hand-containing subfamily further demonstrate how calcium can regulate LIC binding. By combining these data with previous insights from the same group (Lee et al., 2018), the authors provide a complete rulebook for the adaptor-LIC interaction. These data provide strong evidence for a regulatory role for this interaction in cargo recruitment and transport. The data are of high quality and I recommend publication of the manuscript, with some suggestions:

Major Comments

1. The authors show that a mutation in CRACR2a (F58D) results in a reduced numbers of motile dynein complexes, but do not show that this is due specifically to a disruption of its interaction with the LIC. The authors should confirm this with ITC, as was done with BICD2 (Y46D).

Indeed, this was a common request. The mutant was expressed for *in vitro* analysis, and the requested ITC experiment was performed, showing lack of LIC1 binding of the mutant (**Fig. 5a**). Lines 260-262.

2. The authors have compiled an impressive library of LIC1 interactions and binding affinities. Can the authors extract consensus interaction elements that are common to the LIC-binding domains? In line with the spirit of the manuscript, it should be possible to discuss how the conformation and/or identify of the LIC-binding residues correlate with binding affinities. This would show definitively that the interaction is “tunable”, and could even be used as a tool for discovering new adaptors.

This is an interesting suggestion. We now show a superimposition of the three structures of adaptor-LIC1 complexes, using the LIC1 helix as reference for superimposition (**Supplementary Fig. 6**). The superimposition confirms that: a) the conformation and side chain orientations of the LIC1 helix is remarkably conserved, b) there is no specific conservation of the binding interface among different adaptors, other than its overall hydrophobic character. In other words, different adaptors have converged towards this interaction, but using different folds and different contacts. Lines 333-340.

3. The in vitro motility data analysis should be clarified. Assessing the number of motile events per 100 um suggests that all experiments were performed for the same amount of time, but the length of the movies are not stated. It's also not clear if the data points in Figure 5a are from different lysates or from individual microtubules. Clarifications in the legend and methods would suffice.

We addressed the concerns about the in vitro motility data by collecting an entirely new data set using dynamic microtubules to unambiguously assess microtubule polarity and thus ensure that we are scoring only dynein-driven runs. We also more clearly detail our approach in the revised Methods section. Specifically, all videos were 2 min in duration. Each data point in **Fig. 5c** represents the average motility observed along a single microtubule, with at least 5 microtubules analyzed per video and at least 15 analyzed per each replicate. We have performed 3 biological and technical replicates for a final n = 123 microtubules per condition.

Minor Comments

1. In Figure 4a, it might be helpful to include “EFb” somewhere to remind the reader that this structure is different from EFa.

This was apparently confusing. **Fig. 4a** (the structure determined here) comprises both EF-hands (EFa and EFb) of CRACR2a. Indeed, the interaction with the LIC1 helix involves an EF-hand pair, not a single EF-hand. We added labels and use two different shades of orange for EFa and EFb to make this point clear in **Fig. 4a**.

2. In Figure 5a, it might be helpful to include a definition of “MT”.

We added a definition for MT (microtubule) in the legend.

3. In Figure 5b, the y-axis label is “Frequency” but the data is rather a probability. Furthermore, the legend should describe if the data are pooled from different experiments. As another small point here, the authors should consider whether the experiment has the precision of three significant digits, as shown in the mean velocity.

In the current figure (**Fig. 5e**), we present histograms where the y-axis shows the relative frequency of runs of a certain length measured as a percentage. We now state in the legend that the data is pooled from different experiments. We agree with the point about significant figures and have now corrected this in the revised manuscript.

4. The methods for how the “number of motile events” was calculated refers to this as a “landing rate” (line 495), which can be interpreted as including non-motile events. This should be clarified.

We only measured motile events, and this has now been clarified in the corresponding sections of the motility assays. Thank you for pointing this out.

5. In lines 389-390, it's not clear what “as described above” is referring to, given that this is the first mention of protein expression.

We have expanded the description of protein expression (lines 419-424)

6. In Supplementary Figure 1b, it's not clear why some e and g positions are not highlighted; were they intentionally left out?

We have highlighted one additional position that had been omitted (thank you!). Only the positions that support inter-strand salt-bridge interactions are highlighted according to the ZIP+/ZIP- design (Moll et al., *Protein Science* 2001).

7. The authors should state the pixel size of their single molecule data, without which one cannot assess the validity of the 0.5 μm criterion for classifying motile events.

The pixel size is 0.158 μm (added in Methods). This question also led us to reconsider the threshold for motile events; we have now increased the threshold to 1 μm .

8. There is an extra comma on line 390 ("After, ...").

Corrected. Thank you for this and all the other helpful comments.

Reviewer #3

Lee et al. investigate how cargo adaptor proteins activate the motility of dynein, a microtubule-based molecular motor, through their interactions with one of the dynein subunits, the light intermediate chain 1 (LIC1). Dynein transports a large diversity of cargos and this motor does so by binding to many different adaptor proteins, each bound to a specific cargo. The authors focus on the adaptor-dynein LIC1 interaction to understand what commonalities are shared among all the cargo adaptors, which have little sequence homology, and how they may differ to "tune" dynein's transport for specific cellular contexts. By taking a comparative structural approach and using a combination of crystallography, binding assays, and in vitro motility assays, they find that two adaptors exhibit a hydrophobic-binding cleft into which a conserved C-terminal amphipathic helix in LIC1 inserts. The authors also delve into how the adaptor-LIC1 interaction is "tunable" and largely focus on two factors: affinities and Ca^{2+} regulation. This study expands upon their previous work published in *Nature Communications*, which identified the LIC1 helix that binds several cargo adaptors. Overall, this paper effectively expands upon and integrates the information we have thus far on the adaptor-dynein interaction and further demonstrates LIC1's importance in dynein motility. I think the paper will be of interest to those studying dynein and cargo transport, as well as the broad *Nature Communications* readership. However, I recommend that a few issues be addressed before publication.

Major comments:

1. ITC experiments:

The authors focus on the Ca^{2+} binding of CRACR2a, yet in these binding assays, CRACR2a is tagged with MBP. Although a tag is necessary for the detection of CRACR2a (via 280 nm-absorbance), an MBP tag alone is lacking as a negative control. MBP has been shown to bind Ca^{2+} (see Behbehani et al. 2008). Additional data in this study, particularly the structure of CRACR2a co-crystallized with Ca^{2+} , supports that Ca^{2+} does directly bind CRACR2a, however, to obtain an accurate affinity for Ca^{2+} , the authors need to incorporate MBP alone in their ITC assay (Fig. 3).

As stated in the manuscript, we used Maltose-Binding Protein (MBP) tagged CRACR2a in ITC experiments. Behbehani et al. 2008 used Myelin Basic Protein in their experiments. These are

simply two different proteins with the same abbreviation. Nevertheless, for completeness we performed an ITC titration of 800 μM CaCl_2 into 35 μM MBP (these concentrations are higher than those used in our experiments) and detected no binding (**Supplementary Fig. 2a**).

For the ITC experiment, LIC1 (aa 433-458) is purified by reverse phase chromatography in acetonitrile; thus, presumably the LIC1 helix is largely denatured and the affinities measured in this assay may change upon the use of folded protein. Since the varying affinities of cargo adaptors is a large point of discussion in the paper to explain how adaptors tune dynein motility, this caveat should be noted in the text.

Acetonitrile is not present. We state in Methods “*The LIC1₄₃₃₋₄₅₈ peptide was subjected to three cycles of solubilization/lyophilization in methanol 50% (v/v) to remove any trifluoroacetic acid remaining after reverse-phase purification, followed by final solubilization in ITC buffer*”. For clarity, we have added *acetonitrile* to this sentence, any trace of which is also fully removed by this procedure. Both TFA and acetonitrile are highly volatile. Line 447.

2. Single-molecule motility assays:

The authors’ motility data expands upon the study Wang et al. 2019, which showed CRACR2a activates dynein motility and is regulated by Ca^{2+} ; the authors here pinpoint LIC1 as a major contributing factor. However, the wild-type (WT) control used by Lee et al. does not appear to have comparable motility to the WT shown in Wang et al. There are few events of processive motility in the WT control (Fig. 5A). Are the kymographs truly representative of WT motility? There is one notable difference between the two studies: Wang et al. used purified protein, whereas here diluted cell lysate is used and the effective protein concentration in the assay is unclear. Although the authors successfully demonstrate a difference between WT and the F58D CRACR2a mutant, there is little dynamic range in which to explore motility differences since WT already has so few motile events. If WT motility cannot be improved upon by the authors, then the differences between the assays in the two studies should be noted in the text. Also, the y-axis of the graph in A needs to be in “per min per μm microtubule” for easier interpretation.

We agreed with the concerns expressed about our initial data set, and have fully replaced our original data with a more robust data set that (1) uses dynamic microtubules to unambiguously identify minus-end directed runs, which correspond to ~95% of all motile events in our assays; and (2) revised our buffer conditions to ensure sufficient free calcium, following the methods of Wang et al. (2019). As a result, the extent of CRACR2a motility increased substantially. Although the processivity is still lower than that observed by Wang et al., presumably due to their use of purified proteins vs. our use of lysates, which will affect the final adaptor concentration in the assays, the results are now in good agreement. We have also changed the y-axis to “*No. of processive events/min/ μm of MT*” for clarity.

This in vitro data would be nicely complemented with a demonstration of the expected weakened interaction between LIC1 and the CRACR2a F58D mutant—either by ITC or a pulldown—to further support the chosen structure-inspired mutation.

We expressed the F58D mutant of construct MBP-CRACR2a (used in other ITC experiments), and confirmed by ITC that this mutant, used in motility assays, does not bind the LIC1 helix in the presence of Ca^{2+} (**Fig. 5a**). Lines 260-262.

Minor comments:

1. Table 1: Given the BICD2 dataset is anisotropic, the authors should add a few more statistics to be clear about the quality of their dataset. The authors should report the effective resolution

(deff, see Weiss, M. S. Global indicators of X-ray data quality. J Appl Crystallogr, 2001) and the resolution estimates along the three axes, which can be obtained from the anisotropy servers StarAniso or Xtriage. Finally, the redundancy and CC_{1/2} for both crystal structure datasets should be added to Table 1 to further indicate data quality.

We added in Table 1 the overall effective resolution as well as resolution estimates along the three axes. Redundancy and CC_{1/2} statistics were already present in the original version.

2. Lines 102-107: If the structure is reminiscent of the RILP homology-1 domain, then the authors should show this comparison. Otherwise, this seems speculative and tangential.

We describe how the two structures are reminiscent of one another (lines 98-99), not superimposable just reminiscent, and provide a reference to the other structure. We do not believe this side mention deserves a separate figure in a paper that is already quite busy.

3. Fig. 1C: I do not think this is the best way to illustrate the point that A43 and G47 are unusually small and create a cavity. A surface model representation of the binding interface would be more appropriate to show an actual cavity. Alternatively, move the alignment in Supplemental Fig. 1A, which shows the strong conservation of A43 and G47, to the main figure.

We now show a surface representation of BICD2 in Fig. 1e as suggested. Fig. 1c illustrates how the LIC1 helix inserts into the BICD2 coiled coil, where heptad positions *a* and *d*, which are typically bulky and hydrophobic residues, are substituted by Ala and Gly in the CC1 Box to create space for LIC1 residues L444, F447, F448, and L451.

4. Fig. 1E: The interactions should be labeled with the distances to illustrate why Y46 was selected for further study.

Y46 is now highlighted with a different color in Fig. 1e.

5. Fig. 3B: The NIN alignment shows lack of Ca²⁺-binding residues, but FIP3 EFa does not despite being denoted as “non-Ca²⁺-binding” (this is actually noted in the discussion—line 369). Thus, I’m not sure if this section of the figure truly adds anything, certainly not for a main figure and could be moved to the supplement. Even if you look at the Ca²⁺-binding residues in the true Ca²⁺-binding EF hands, not all of these residues highlighted as highly conserved are actually conserved, making it feasible that the FIP3 EFa domain may bind Ca²⁺ (perhaps upon binding an accessory protein). Note also, that one of the Ca²⁺-binding aspartic acids is conserved in FIP3 EFa but not highlighted purple to indicate conservation.

We have experimentally shown how CRACR2a binds Ca²⁺ whereas FIP3 and NIN do not (Fig. 3a), and it is good to have this experiment and the alignment (Fig. 3b) in the same figure. To understand why certain amino acids are allowed at specific positions of the Ca²⁺-binding loop and others are not, it is important to look at the structures (Fig. 4d). For instance, positions Y and Z can be D, N or S because these positions contribute a single side-chain oxygen to the interaction with Ca²⁺, and accordingly all three residues are found at this position in the “Ca²⁺-binding EF-hands” group. Similarly, position -X contributes a main chain contact, and thus any amino acid is allowed at this position. On the other hand, we specifically explain why D at position -Z (as observed in FIP3 EFa), which makes a bidentate interaction with Ca²⁺, is not allowed. We further cite a paper that specifically studied this question (Cate et al., *Structure* 1999) and address this potentially confusing point in the Discussion. While we cannot further discuss the many nuances of the EF-hand loop in this paper, we have added some clarifications

in the text (Lines 233-239). We also labeled positions X to -Z in **Fig. 4** as in **Fig. 3b**. For the interested reader we also cite reviews and references that further expand upon this well-studied topic.

6. Fig. 4D: I do not think it is necessary to show both troponin and calmodulin in the figure to show CRACR2a is binding Ca²⁺ similarly; perhaps only one should be included. If there is good reason for showing troponin and calmodulin, the authors should state this clearly in the text.

The question above (point 5) illustrates why it is a good idea to show both troponin and calmodulin. We have added sentences in the main text to clarify how the X to -Z positions of the EF-hand loop contribute differently to the binding of Ca²⁺, and how these can (and cannot) vary in prototypical Ca²⁺-binding loops (as exemplified by troponin and calmodulin, that while different both bind Ca²⁺).

7. Line 249: The authors should state the rationale for why F58 was mutated to specifically aspartic acid and not a different amino acid, such as alanine.

Like with Y46D in BICD2, the goal of the F58D mutation in CRACR2a was to replace a large aromatic side chain with a negatively charged side chain that conflicts with hydrophobic residues in the LIC1 helix. We have added an ITC experiment showing the inability of the mutant to bind the LIC1 helix and a close-up view of F58D (**Fig. 5a**), along with requested clarifications in the text (Lines 260-262).

8. Many figure legends are lacking the number of times the experiment was conducted. Authors should include “n =” to indicate reproducibility.

If this comment refers to ITC titrations, n=1. After finding appropriate conditions, which may require multiple trials, the ITC experiments are performed once. Yet, each titration implicitly consists of multiple experiments. This is always the case with published ITC data. The same applies to crystal structures, which are determined once (if multiple crystals are analyzed we commonly obtain the same structure, even if the crystallization conditions change). In **Fig. 5**, we performed 3 biological and technical replicates for a final n = 123 microtubules per condition, and this is indicated.

9. The discussion can be improved. Paragraphs 2-3 should be condensed because as written, it comes across as a review and is not directly put in the context of the current study. Also, a number of broad implications can be derived from this study, yet the last paragraph does not do this justice. The discussion should end on the wider implications at hand, rather than a focus on proteins that lack data to show they are true dynein-dynactin adaptors.

There is indeed some flavor of a Review in the Discussion. We have now determined the three main structures that characterize the LIC1-adaptor interaction, and this needed to be put into a general context for the reader (as pointed out in point 10 below). We describe how these three structures separate the known Adaptors into three subfamilies. We also try to place these findings within the context of existing cryo-EM structures. We have reworded sentences during revisions that address some of the new data and questions asked by the reviewers.

10. Fig. 6 and Supplemental table 1 are great additions to the study and will be useful for the dynein community.

Thank you for the helpful evaluation of the work!

REVIEWERS' COMMENTS

Reviewer #1 (Remarks to the Author):

The authors did a fantastic job of addressing my concerns about their original submission. I have no additional comments and am excited to see this paper get published.

Reviewer #2 (Remarks to the Author):

The authors have done a great job with additional controls and figure/text clarifications. The reworking of the TIRF assay strengthens the manuscript, but I have a few concerns with it:

- The microtubules are not technically polarity marked since microtubules can grow from the minus ends of the seeds. Therefore, motile events towards the seed would be growing towards the plus end. The authors should clarify their criteria for making sure they were looking primarily at growing plus ends (e.g. the faster/longer of the two growing ends, etc.).
- There number of processive events in the F58D is non-zero, as revealed by the numerous data points in panels c, d, and e. However, the kymograph shown in panel b does not show any processive events. The authors should include kymographs of F58D showing processive events.
- Run lengths should analysed by fitting an exponential decay to the cumulative distribution function, from which one can extract a decay constant (i.e. mean run length).
- The x-axis for the run length plot is slightly misleading and should either start at 0 or label the first tick to make it clear it starts at 1.
- The tubulin concentration in the experiment is not stated.

Reviewer #3 (Remarks to the Author):

The authors addressed all major concerns. The in vitro motility data was particularly improved, and all requested controls were done. I am in full support of publication given the current revision.

ADDITIONAL COMMENTS (REVIEWER-2)

We thank the three reviewers for their helpful comments. We address here additional minor issues raised by reviewer-2. Changes are highlighted in the manuscript. In the response below, we use black text for the reviewers' comments and blue text for the authors' response.

Reviewer #1

The authors did a fantastic job of addressing my concerns about their original submission. I have no additional comments and am excited to see this paper get published.

Thank you for the helpful evaluation of the work.

Reviewer #2

The authors have done a great job with additional controls and figure/text clarifications. The reworking of the TIRF assay strengthens the manuscript, but I have a few concerns with it:

1. The microtubules are not technically polarity marked since microtubules can grow from the minus ends of the seeds. Therefore, motile events towards the seed would be growing towards the plus end. The authors should clarify their criteria for making sure they were looking primarily at growing plus ends (e.g. the faster/longer of the two growing ends, etc.).

We did as suggested above, considering only microtubules that had one end growing clearly faster (+ end) than the other (- end). Only movements toward the slow growing end of these microtubules were analyzed. This was clarified in the Methods.

2 There number of processive events in the F58D is non-zero, as revealed by the numerous data points in panels c, d, and e. However, the kymograph shown in panel b does not show any processive events. The authors should include kymographs of F58D showing processive events.

We have included one example of a kymograph showing a motile event with F58D in the presence of Ca^{2+} (**Supplementary Figure 4e**).

3 Run lengths should analysed by fitting an exponential decay to the cumulative distribution function, from which one can extract a decay constant (i.e. mean run length).

Thank you for this suggestion. We now fit the histograms to an exponential decay and report the R^2 and curve equation ($y = y_0e^{-t}$) in Fig. 5e.

4. The x-axis for the run length plot is slightly misleading and should either start at 0 or label the first tick to make it clear it starts at 1.

Indeed, the graph starts at 1 (a label was added as suggested).

5. The tubulin concentration in the experiment is not stated.

The MT seed (250 nM) and free tubulin (11.25 μM) concentrations have been added in Methods

Reviewer #3

The authors addressed all major concerns. The in vitro motility data was particularly improved, and all requested controls were done. I am in full support of publication given the current revision.

Thank you for the helpful evaluation of the work.